# Social subordination is associated with better cognitive performance and higher theta coherence of the mPFC-vHPC circuit in male rats

**Faezeh Zarfsaz¹, Soomaayeh Heysieattalab[ID]¹\*, Ali Jaafari suha²,
Farhad Farkhondeh Tale Navi[ID]¹, Hamid Basiryan¹**

**1** Department of Cognitive Neuroscience, Faculty of Education and Psychology, University of Tabriz, Tabriz, Iran, **2** Department of Physiology, School of Medicine, Shahid Beheshti University of Medical Sciences, Tehran, Iran

\* heysieattalab@gmail.com

## Abstract

Social dominance hierarchy is considered an influential factor on cognitive performance. The spatial working memory (SWM) is inversely related to dominance status after the formation of social hierarchy. However, their neural underpinings are poorly understood. The medial prefrontal cortex (mPFC) and ventral hippocampus (vHPC) play pivotal roles in social hierarchy and SWM. To investigate the associations between social hierarchy and SWM and their neural circuit (mPFC-vHPC), we used twenty one natal male Wistar rats after weaning (3 rats per cage, 7 cages in total). In the 9th postnatal week, the tube test was started to determine the relative social rank in each cage (dominant, middle-ranked, subordinate). One month after living in the hierarchy, we implanted electrodes in mPFC and vHPC. One week following recovery, the SWM test was performed using T-maze with two difficulty levels (30s and 5min delays between trials) while recording the local field potentials. The percentage of correct responses showed no significant difference among three different social groups. However, subordinates demonstrated significantly lower latency in reaching the goal arm, while middle-ranked rats exhibited the longest latency in 30s delay. Electrophysiological data revealed significantly higher theta correlation and coherence of the mPFC-vHPC circuit in subordinates. Although theta rhythm synchronization was reduced in all social ranks by increasing task difficulty, the subordinates maintained better task performance and less reduction of theta coherence. These findings underscore the association between social hierarchy and working memory performance within the mPFC-vHPC circuit, highlighting the influence of social rank on implicated circuit.

## 1. Introduction

Social cognition, encompasses the intricate processes by which we perceive, retain, and utilize information to understand and predict behavior within societies [1]. At its core, social cognition deals with the various social cues that enable adaptive responses to the the surrounding

**Data availability statement:** Data files are available within the OSF database: https://osf.io/sjex3/?view_only=c0da26c-dab6e4141995620a703a016d8

**Funding:** We would thank the Iran National Science Foundation for foundation support (grant number: 4012752). The funders had no role in study design, data collection and analysis, decision to publish, or preparation of the manuscript."

**Competing interests:** The authors have declared that no competing interests exist.

social environment [2]. Among the myriad of social interactions facilitated by our brains, the establishment of social hierarchies stands out as a fundamental aspect [3].These hierarchies serve as predictors for physical well-being, cognitive functions, and in extreme cases, the survival of organisms [4,5], essentially acting as organizational frameworks for the allocation of limited resources such as mates and food [6].

Research has established that cognitive processes, particularly working memory, are influenced within the social context. Spatial working memory (SWM), a pivotal subtype of working memory assumes a critical role in encoding threats, resources, and environmental cues, thereby facilitating adaptive decision-making and interactions [7–9].The indispensability of SWM in purposeful behavior underscores its relevance in navigating dynamic social environments [9]. Considering the interdependence of social and cognitive functions, it becomes pertinent to explore the reciprocal relationship between social hierarchy and SWM. Does possessing heightened cognitive abilities confer higher social status, or does social standing precipitate alterations in cognitive functions. In other words, do social life and its responsibilities cause changes in cognitive abilities, or is it the other way around?

The medial prefrontal cortex (mPFC) and ventral hippocampus (vHPC), along with their intricate interactions, emerge as central players in higher cognitive processes including memory and learning [10]. The PFC, particularly implicated in executive control of behavior, assumes a primary role in processing information pertinent to the social hierarchy [11]. Within the realm of social competition, the medial prefrontal cortex (mPFC) is a key area mediating the formation of social hierarchies [12,13]. Complementing the role of the mPFC, the vHPC assumes significance in social memory, encompassing the neural representation of individuals and their social statuses [14]. While the precise involvement of the vHPC in social hierarchy remains to be fully elucidated, emerging evidence suggests its pivotal role in encoding and recalling episodic social information [15]. Moreover, the intricate interplay between the mPFC and vHPC extends to their joint involvement in SWM tasks. The successful execution of SWM tasks relies on the coordinated activity of these regions, with the ventral hippocampus (vHPC) and dorsal hippocampus (dHPC) contributing to spatial encoding, and the mPFC facilitating executive control [16]. Notably, the synchronization of theta rhythms between the mPFC and vHPC fosters long-range connectivity crucial for SWM formation [17,18]. In this study, we used vHPC because the synchronization between PFC and vHPC is stronger than dHPC. Also, inputs from vHPC is very important for mPFC and plays a significant role in synchronizing the oscillatory activity of mPFC with dHPC area. In other words, theta rhythm in the vHPC regulates the synchrony between these two regions [19,20].

Several studies have shown discrepancies in the SWM performance across animals with varying social statuses. While some studies indicate superior SWM performance among dominants, others report contrasting findings [21,22]. Recently, in a study conducted on home-caged sibling laboratory rats with various intervening variables being controlled, it was shown that the performance of SWM in the Morris water maze (MWM) in subordinate rats after forming a dominance hierarchy is significantly better than the rats with higher social status. However, in this study, SWM in the Y-maze task did not show significant difference between dominants and subordinates, unlike the MWM [10]. As the authors have discussed, one possible explanation for discrepancies between the results of MWM and Y-maze regarding dominance hierarchy could be the varying cognitive load of each task. The cognitive load in working memory tests is mainly determined by the delay period between finding a cue and responding after a delay for which increasing the delay period will increase the cognitive load [23]. As recently has been shown, the intact mPFC and HPC and their synchronous activity are essential for the successful performance of the SWM task with higher cognitive load [24]. This phenomenon underscores the pivotal role of the mPFC-HPC circuit in mediating SWM

under varying cognitive demands. Increasing the degree of task difficulty means increasing the delay time between the sample and the choice phases of the SWM task. It seems as the difficulty of the task is increased, the integrated activity between the mPFC and HPC increases to produce more accurate spatial representation [10]. The increased connection between these two areas is reflected as an increase in the firing rate of neurons and the synchronicity of their activity [25]. The results have shown that in longer delays between the sample and the choice phases, the interaction of the mPFC and the HPC is necessary to coordinate the memory processes related to the past and the future in predicting the achievement of a goal; while at short delays, each of the two involved structures mentioned above independently represents sufficient spatial information to succeed in the task [26]. Considering controversial results from different studies regarding relationship between social hierarchy and memory performance [10,21,22], our study tries for the first time to elucidate the impact of social hierarchy on SWM during two degrees of task difficulty, specifically aiming at shedding light on the underlying functional connectivity within the mPFC-vHPC circuit by LFP recording in sibling male rats with different social ranks.

## 2. Materials and methods

### 2.1. Animals and ethical statement

Twenty-one male Wistar rats were used in this study. They were home-caged postweaning as triads of litters from the same mother with similar weight (70–80 g). Monitoring of weight started from birth and continued weekly until the surgical interventions. The rats were obtained from Urmia Medical Sciences University and housed at 21±2 °C, 12:12 light-dark cycle (from 8 AM to 8 PM). They were kept in standard animal research facilities in which food and water were freely available. Ethical approval for the study protocol was approved by the "Research Ethics Committee of Tabriz University (IR.TABRIZU.REC.1401.022)", in compliance with established guidelines for the ethical treatment of animals in research.

### 2.2. Study design

This research was conducted in two behavioral and electrophysiological phases. In the behavioral phase, the triads underwent tube test training in the 9th postnatal week and the tube test was performed during weeks 10–12 to determine their relative social rank in each cage. After living in the social hierarchy for one month, the surgery was performed at 18–24 weeks of age and, after seven days of recovery, the SWM test with electrophysiological recording was performed between weeks 18–24. To evaluate the SWM, the spontaneous alternation in the T-maze test was used without any rewards. The reason for choosing this protocol is the possible difference in desire for food between dominant and subordinate rats [27]. Also, in the T-maze test, the delay time between the sample and the choice phases was alternated by different levels of difficulty (easy: 30s and difficult: 5min delay time between sample and choice run trials). The cognitive load of the task increases by increasing the difficulty level of the SWM task. So, it was presumed to better distinguish between the cognitive performance of dominants and subordinates. The electrophysiological phase was performed along with the SWM task. At this stage, the changes in neuronal activities of the vHPC-mPFC circuit after the formation of social hierarchy were investigated during the SWM task in T-maze (Fig 1).

2.2.1. Behavioral tests. 2.2.1.1. Social dominance tube test; The social dominance tube test is a standard test that used to assess the relative dominance status among triads of each cage. It is made up of a transparent Plexiglas tube, 1 m in length and 5.4 cm in inside diameter, allowing passage of a single adult rat without the possibility of reversing its direction inside the tube [10]. A plastic box was connected to either end of the tube to facilitate the

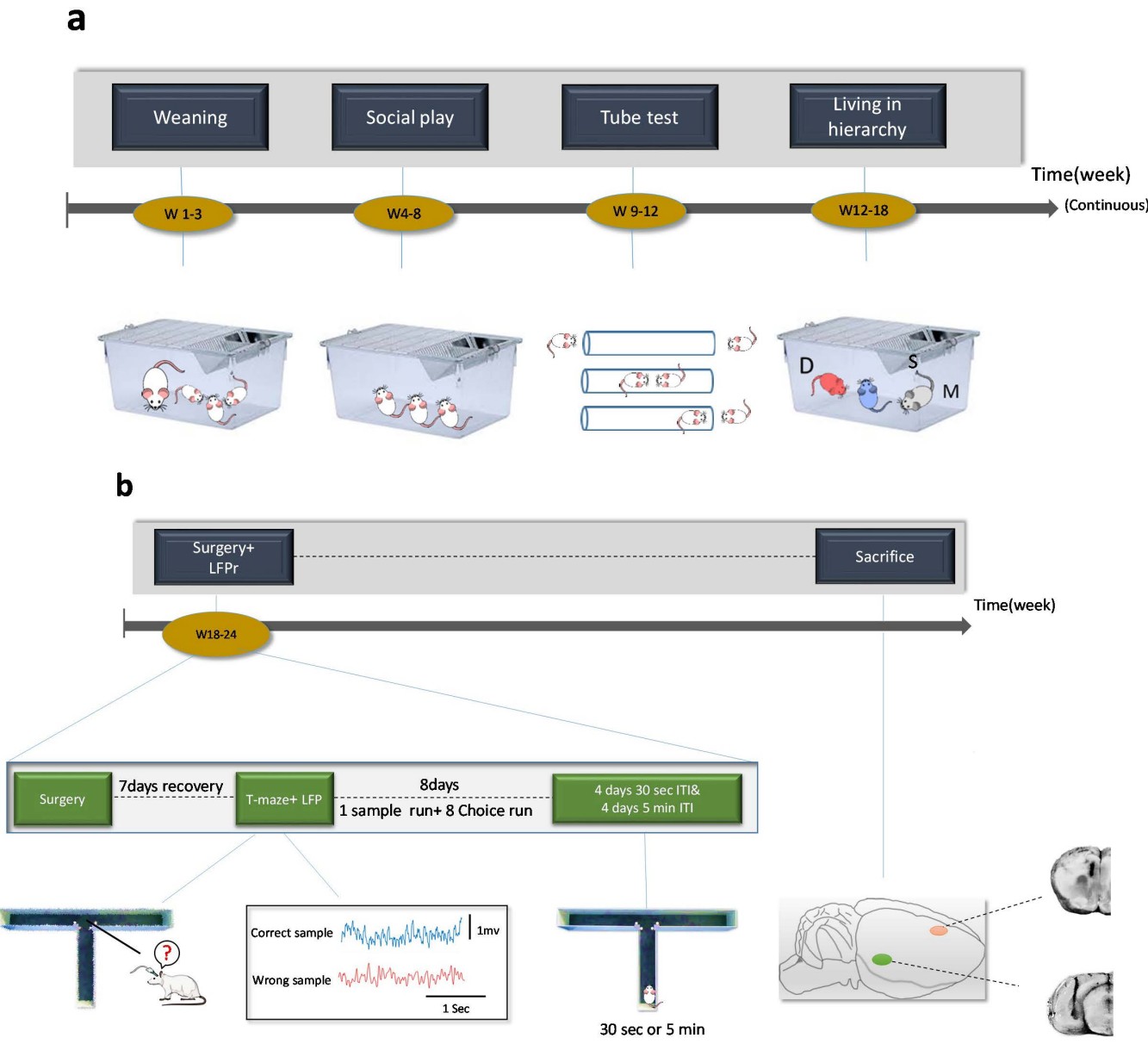

**Fig 1. Timeline of behavioral and electrophysiological procedures.** a) Different phases of social development of three natal littermates before and after tube test: Dominant (D, red), Middle-ranked (M, gray), and Subordinate (S, blue) b) Timeline of surgery, T-maze task, LFP recording in decision point (4 sec) with the correct and wrong responses, and histological confirmation of the sites of the electrodes in mPFC and vHPC (LFP: local field potential; mPFC: medial prefrontal cortex; vHPC: ventral hippocampus).

entry and exit of the rats. Before starting the tube test, rats underwent daily handling for one week to mitigate anxiety. In the ninth postnatal week, tube test training was performed for three days during which each rat traversed through the tube eight times, alternating the entry side between the left and right side of the tube for each passage. After training, the tube test round-robin tournament was conducted in five test days during postnatal weeks 10–12. During each test trial, each pair of rats entered the tube simultaneously from opposite ends of the tube and met each other in the middle of it. One of the rats that forced other rat to retreat with four paws out of the tube gained one score and the retreated animal gained zero. This

was repeated by changing the entry end for rats on each test trial. There were three test trials for every cage in each test day that every rat contested the other two rats (e.g., A-B, A-C, and B-C). Averaging the number of wins for five test days determined the relative dominance status of rats in each cage in descending order as dominant (D), middle-ranked (M), and subordinate (S) [3,12]. Once there was no winner for more than 2 minutes or both of the rats retreated out of the tube, the test was repeated. Moreover, when changing the entry side resulted in a different outcome, the test was repeated until one of the rats became the winner or each rat gained 0.5 score. This test was done in the afternoon every dayfrom 3pm to 6 pm. The tube was cleaned with 70% ethanol after each test trial. Also, the light intensity during the experiment was 100 lux.

**2.2.1.2. Spontaneous alternation in T-maze test;** T-maze is a widely employed apparatus (nominated due to its similarity to the letter T) in cognitive experiments in animals that offers a simple choice between the left arm and right arm. T-maze is used to study memory function and spatial learning through the application of different stimuli. According to the standard protocol for T-maze running [23], we used the spontaneous alternation which is performed without a food reward and habituation. Here, it is the novelty of the maze arms that drives the spontaneous exploration. We used this protocol because of rats difference in desire for food between dominant and subordinate rats [27]. In T-maze task, for the sample phase, each rat was first placed at the starting point (i.e., the base of the arm perpendicular to the left and right arms) of the maze and was let to choose between left or right arms. After the sample phase, the rat was locked in the selected arm for 30 seconds and then returned to the starting point. At the starting point, depending on the difficulty of the task, rats were locked at that point for 30 s (easy) or 5 min (difficult), a delay time between sample and choice run trials. In this protocol, it is possible to freely choose the target arm in both the sample phase and choice phase [23]. The difficulty of the task was alternated every day for eight days (four days of 30 seconds difficulty and four days of 5 minutes). Six trials were performed every day (one sample trial and five choice trials). The time limit for the rats to remain in the maze was 180 seconds [23]. All movements of the animals were recorded via a video camera located above the maze for subsequent analysis. All tests were conducted in the afternoon every day from 3 pm to 6 pm. Also, the light intensity during the experiment was 100 lux (Fig 1b).

**2.2.2. Electrode implantation and histological verification.** To implant the electrodes in the specified areas, at first the rat was anesthetized using a mixture of ketamine (100 mg/kg) and xylazine (10 mg/ kg). Then, the anesthetized rat was placed in the stereotaxic apparatus (Toosbioresearch, Mashhad, Iran) and a heating pad was used to maintain the rat body temperature at 37 °C during surgery. Anesthesia depth was checked by tail and foot pinch reflexes. Vitamin A ointment was also used to prevent the animal's eyes from drying out during surgery. After drilling the skull, stainless-steel electrodes (127 μm in diameter, A.M. System Inc., USA) were implanted unilaterally into stereotaxic coordinates of mPFC (AP:+3.2 mm; ML: −0.6 mm; DV:−3.6 mm) and vHPC (AP: −4.92 mm; ML: −5.5 mm; DV: −7.5 mm) according to the rat brain atlas [28]. One additional hole was drilled into the skull and used as the reference electrode. Then we poured dental adhesive on the skull and placed the socket on it.

To ensure accurate electrode placement, rats were first deeply anesthetized with carbon dioxide and then the brains of the rats were carefully removed and placed in 4% paraformaldehyde at 4°C for 48h. Then, using a vibroslicer, the brains were cut and stained with methylene blue and examined under a microscope (AC 230V 50 Hz, Fig 1b).

**2.2.2.1. Electrophysiological recordings;** After implanting electrodes and one one-week recovery period, local field potential (LFP) signals were recorded from the mPFC and vHPC areas. The recording method included recording brain signals from each rat for eight testing

days. Due to the spontaneous selection of the SWM task, the rats did not need to be trained and habituated to the maze space. Thirty min before the start of the recording, the rats were transferred from the animal room to the testing room to habituate the room and the recording chamber (isolated from sound and environmental noises). Then, they were placed at the starting point of the maze, and recording was conducted during the test. The important point that was considered for review and analysis was the decision point (the moment of choosing one of the left or right arms plus three seconds before and one second after this point; overall 4 seconds). The LFP recordings (BIODAC-A, TRITA Health Tec., Tehran, Iran) were made through the digital head stage probe, which can simultaneously record five brain regions and a two-channel amplifier. In the frequency settings, waves below 250 Hz were selected for recording. During the test, all the steps were recorded using the camera placed inside the chamber. Below are pictures of the steps of the test as well as the comparison of the raw data of correct and wrong responses (Fig 1b).

## 2.3. Data analysis

The process of extracting the data from the electrophysiology section was done by using videos recorded during the LFP signal recording, in order to match the time of behavioral activity with LFP signals. Then, the videos were analyzed and the decision point was extracted with millisecond precision. This time is actually the moment of turning the rat's head towards the target for the last time, which three seconds before this time and one second after this moment (4 seconds in total) was considered as the decision point. Then these files were converted into readable files for MATLAB version 2022b (MathWorks, Natick, MA, USA), and the desired times were extracted from the recorded signals. In this way, theta distribution power, theta correlation between mPFC and vHPC regions, and theta coherence between these two regions were extracted in the frequency range of 4–12 Hz at the decision point of the rat to choose the desired arm. The desired variables of this research were analyzed using functions available in MATLAB such as "mscohere" function for circuit coherence. Results represent the cumulative number of correct and incorrect responses across the four days of trials; because the difficulty of the task was alternated every day for eight days (four days of 30 seconds difficulty and four days of 5 minutes difficulty). Also, LFP analyses conducted for each individual trial.

Finally, for statistical analysis, the numerical data extracted from MATLAB was transferred to GraphPad Prism 9.5.0 software and statistical analysis was performed using two-way analysis of variance and post hoc tests. To extract behavioral data, the percentage of correct responses and the time taken by the rat to select the goal arm (latency) were extracted by meticulously analyzing recorded videos of behavioral sessions. Finally, for statistical analysis, the data was transferred to GraphPad Prism 9.5.0 software, and statistical analysis was performed using two-way analysis of variance and post hoc tests.

## 3. Results

### 3.1. Behavioral assessments

**3.1.1. Social dominance tube test.** A test tube was used to determine the relative social dominance of rats in each cage (N = 7 cages, 3 rats/cage). To determine the final rank of each rat in each cage, the number of wins for five test days were averaged and assigned in descending order as dominant (D; rank 1), middle-ranked (M; rank 2), and subordinate (S; rank 3), respectively. One-way ANOVA revealed significant difference in the mean number of wins between rats with different social rank ($F_{(2, 102)} = 139.2$, *P<.001*). Post-hoc analysis using Bonferroni test indicated significant difference between D and M (*P* <.001), D and S

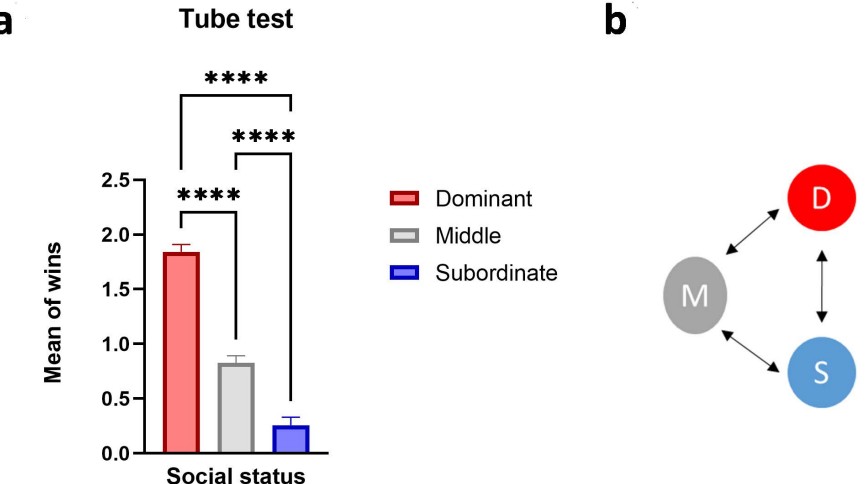

**Fig 2. Determining relative dominance using tube-test.** a) Schematic dominance relationship between triads of dominant (D), middle-ranked (M), and subordinate (S) rats. b) The mean number of wins of each social status. Values express as the mean ±SEM.,****p<.0001.

(*P*<.001), and M and S (*P*<.001) (D: 1.843±.06727, M:.8286±.06463, S:.2571±.07210for all cages; Fig 2).

**3.1.2. The percentage of correct responses in T-maze with two difficulty levels.** The percentage of correct responses in the choice run during *easy task* (30 s delay) was determined for each session in three groups based on the T-maze task performance (Fig 3a). One-way ANOVA revealed no significant difference between rats with different dominance status (F (2, 18) =.5266, *P*=.5995; D: 60.00± 8.797; M: 52.86± 9.503, and S: 66.43± 4.253; Fig 3 b, left panel).

Using one-way ANOVA to compare the percentage of correct responses in the choice run during *the difficult task* (5 min delay) again showed no significant difference between three groups (F (2, 18) =.7786, *P*=.4739; D: 59.29± 9.785; M: 48.57± 9.044, and S: 65.00± 9.512; Fig 3 b, middle panel).

Additionally, we used the two-way ANOVA to compare the percentage of correct responses between *two difficulty* levels for each rank. There was no significant main effect of task difficulty (F (1, 36) =.07788, *P*=.9800), and interaction effect (F (2, 36) =.02019, *P*=.9800; Fig 3 b, right panel).

**3.1.3. The latency of responses in T-maze with two difficulty levels.** Comparing the latency of correct responses in the *easy task* between rats with different dominance status using the one-way ANOVA exhibited a statistically significant difference (F (2, 18) = 21.82, *P*<.001). Bonferroni's multiple comparisons test revealed a significant difference only between subordinates and dominants (t (18) =4.215, *P*<.001), and subordinates and middle-ranked rats (t (18) =6.512, *P*<.001). These findings show that the subordinates had the shortest latency, while the middle-ranked rats had the longest latency during all sessions (D: 69.16± 3.085; M: 80.24± 3.607, and S: 48.84± 3.513; Fig 3 c, left panel).

Analysing the latency in the *difficult task* between three groups using one-way ANOVA showed no significant difference (F (2, 18) =.7709, *P*=.4773). These findings show that animals with different social rank have similar delay time to choose the target arm in difficult task (D: 77.29± 7.504; M: 80.37± 7.136, and S: 69.33± 4.346;Fig 3 c, middle panel).

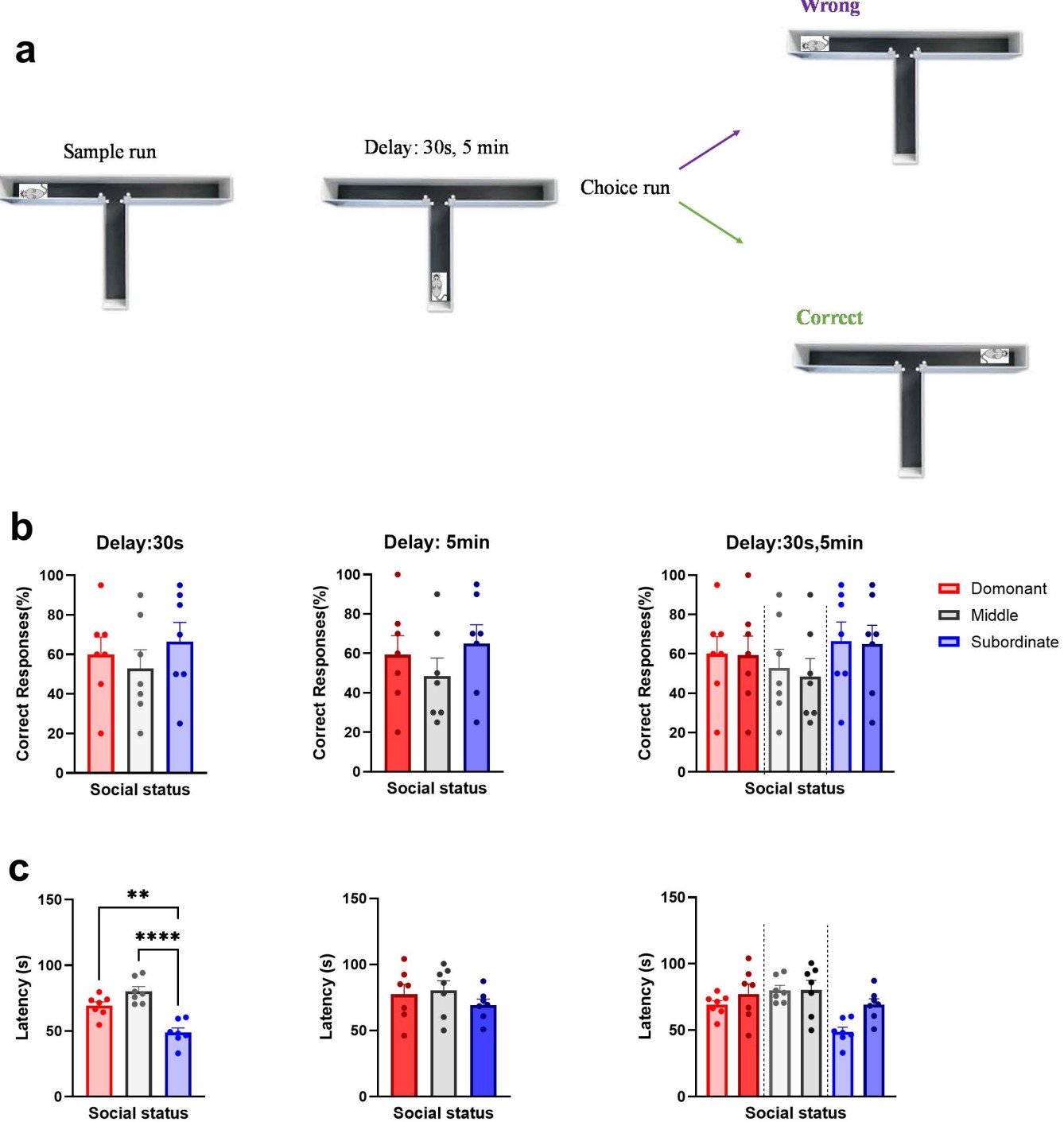

**Fig 3. Performance in T-maze task with two difficulty levels** a) T-maze spontaneous alternation test comparing b) The percentage of correct responses in easy task (30s, left panel,) and difficult task (5 min, middle panel) and comparison of two difficulty levels for each rank (right panel). c) Comparing the latency in easy task (30 sec, left panel; **p<.01,***p<.001) and difficult (5 min, middle panel) and comparison of two difficulty level for each rank (right panel). Values express as the mean ±SEM.

Further, comparison of the latency of the correct responses between *two difficulty* levels using two-way ANOVA demonstrated difference in main effect of task difficulty (F (1, 36) = 5.133, *P=.0296),* but not in interaction (F (2, 36) = 1.962, *P=.1553*, Fig 3 c, right panel).

## 3.2. Electrophysiological assessments

### 3.2.1. Theta rhythm power of the mPFC during responses in T-maze with two difficulty levels.

We investigated changes in mPFC theta power among three groups of dominant, middle-ranked, and subordinate rats during correct and wrong responses in two difficulty levels of the T-maze task. Examples of the time-frequency spectrograms and raw LFP of mPFC in the subordinate group are shown in both degrees of task difficulty during correct and wrong responses (Fig 4 a b).

Two-way ANOVA was used to compare the theta power of the mPFC in the correct and wrong responses in three social rank groups during *easy task.* Comparison of *correct and wrong* responses for each social rank showed significant group main effect (F (2, 162) = 3.964, *P=.0209*). Moreover, comparing the response type revealed significant differences between groups (F (1, 162) = 76.28, *P<.001*). Significant interaction effect also was revealed (F (2, 162)

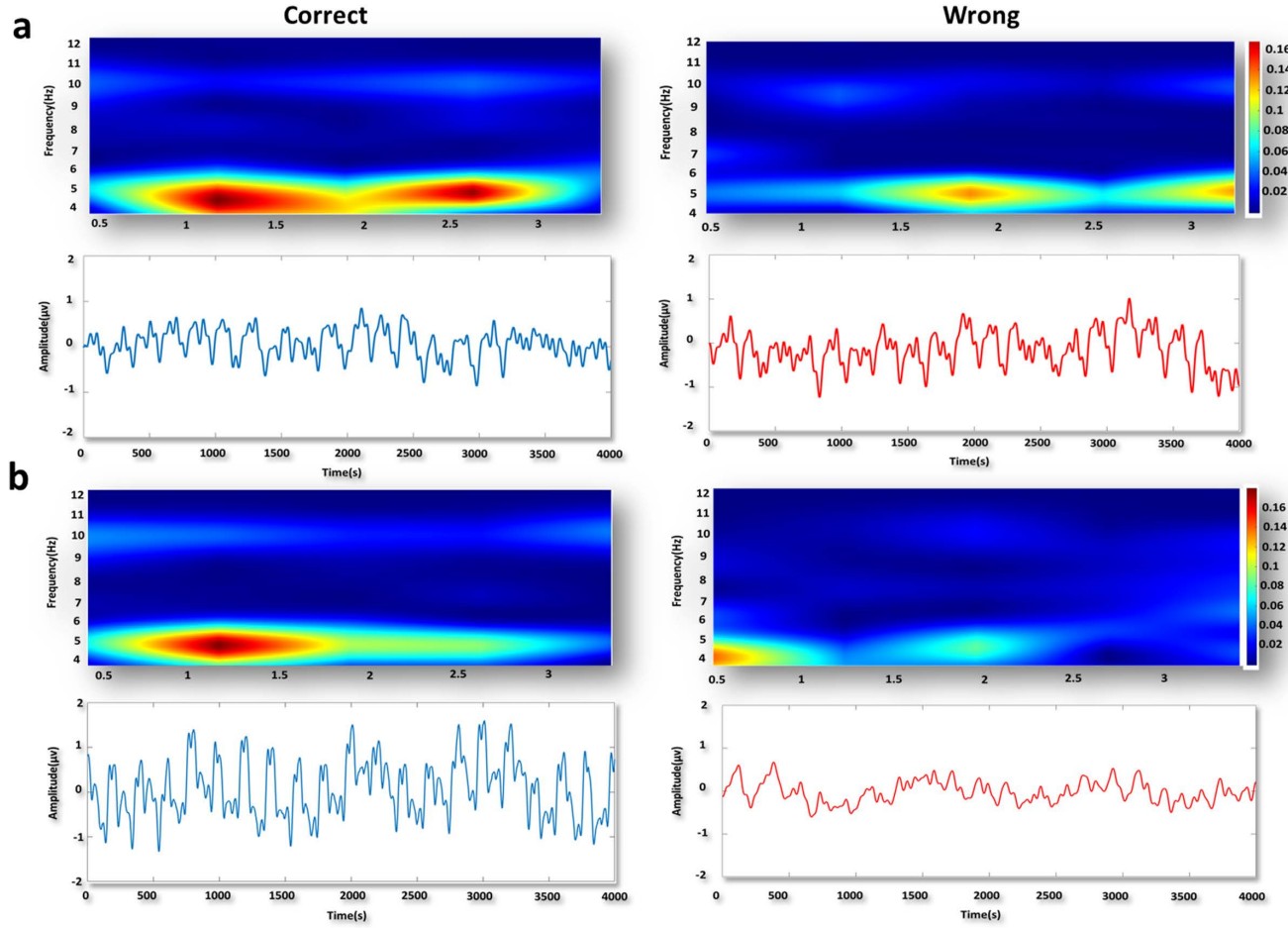

**Fig 4. Sample time-frequency spectrogram and raw LFP signals in mPFC in subordinates.** a) Sample time-frequency spectrogram of one correct trial and its raw signal (blue) and the wrong one (red) in mPFC in easy task (delay: 30 s). b) Sample time-frequency spectrogram of one correct trial and its raw signal (blue) and the wrong one (red) in mPFC in difficult task (delay: 5 min).

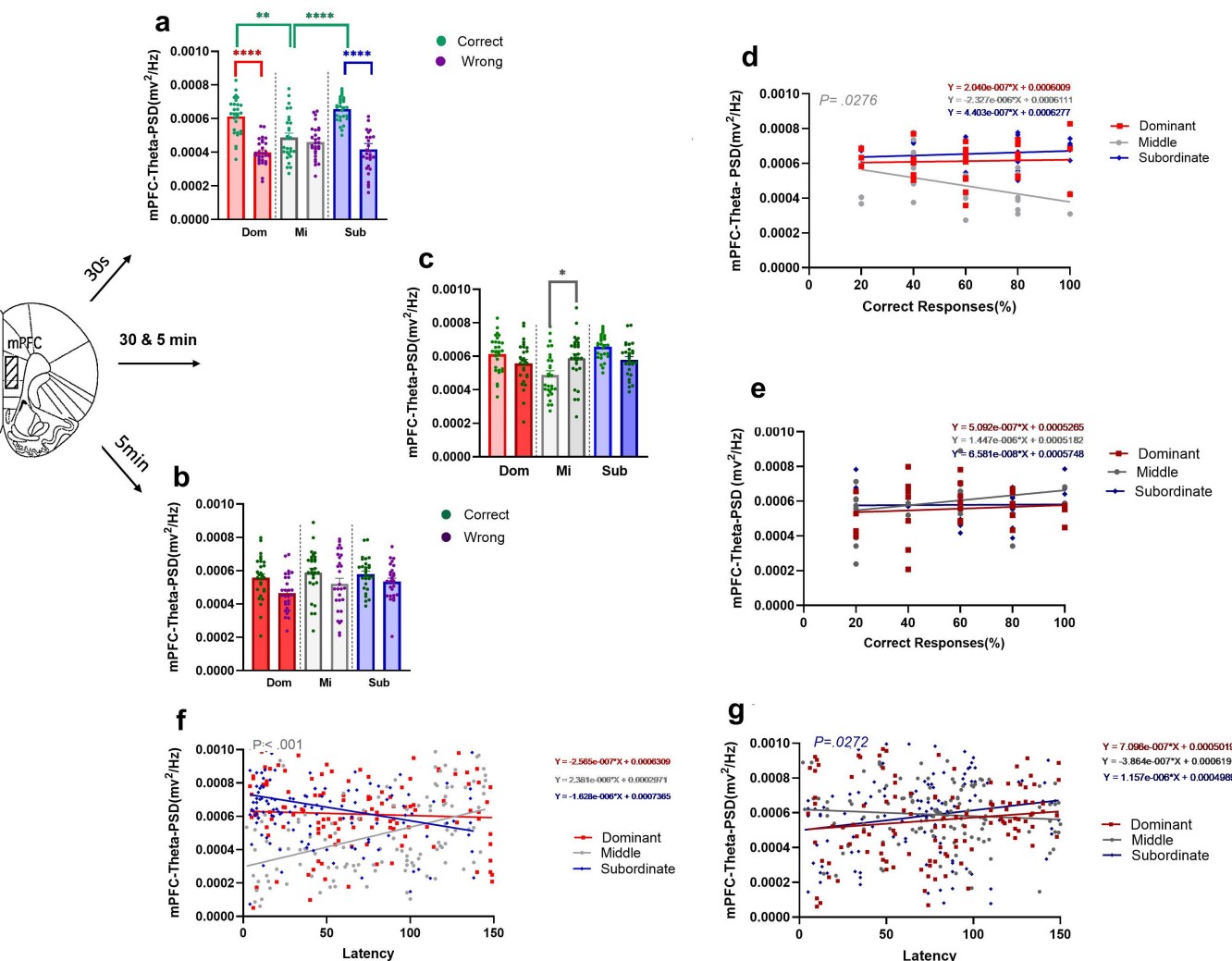

**Fig 5. Mean theta rhythm power (PSDs) of mPFC in T-maze task with two difficulty levels in dominant, middle-ranked, and subordinate animals.** The mean PSDs of mPFC in correct and wrong responses during (a) easy (* p<0.05, ** p<0.01, **** p<0.0001; two-way ANOVA) and (b) difficult tasks. **(c)** Comparing PSDs of two difficulty levels for each social rank in correct responses (two-way ANOVA). Values are expressed as mean ± SEM. Correlation of mPFC theta rhythm power with the percentage of correct responses during (d) *easy* (M: r= -.4161) and (e) *difficult* tasks of three social ranks for 4 days. Correlation of mPFC theta rhythm power with response latency during (f) *easy* (M: **r** =.4514) and (g) *difficult* (S: r=.2017)tasks of three social ranks for 4 days. mPFC: medial prefrontal cortex, PSD: power spectrum density.

= 13.12, *P<.001*; Fig 5a). Bonferroni's multiple comparisons showed a significant difference in *correct responses* only between dominant and middle (t (162) = 3.930, *P* =.0019) and subordinate and middle (t (162) = 5.309, *P* <.001; Fig 5a) groups. Comparing between correct and wrong responses together using Bonferroni's multiple comparisons test showed significant differences only in dominants (t (162) = 6.721, *P<.001*) and subordinates (t (162) = 7.521, *P<.001*; Fig 5a). So,dominant and subordinate rats showed higher theta power of mPFC in the *easy task*.

When we compared the theta power of the mPFC in *correct and wrong responses* for each social rank during *difficult task* using two-way ANOVA, while there was no significant differences was found in group main effect (F (2, 162) = 2.167, *P=.1178*), but the main effect of response type was significant (F (1, 162) = 11.08, *P=.0011*). Furthermore, there was no significant interaction effect (F (2, 162) =.4426, *P=.6431*; Fig 5b).

Furthermore, using two-way ANOVA to compare the theta power of the mPFC in *correct* responses between *two difficulty levels* for three social ranks was found no significant main effect of *task difficulty* (F (1, 162) =.3802, *P=.5383*; Fig 5c). However, interaction effect was significant (F (2, 162) = 9.409, *P<.001*). Bonferroni's multiple comparisons test showed significant differences between dificulty leveles in middle (t (162) = 3.164, *P=.0279*)

**3.2.2. The correlation between theta rhythm power in the mPFC and the percentage of correct responses in T-maze with two difficulty levels.** Correlation analysis between the theta rhythm power of the mPFC and the percentage of correct responses during the *easy task* showed that with the increase of theta rhythm power, the percentage of correct responses increased in subordinate and dominant rats (Dominant: *r=.04456, P=.8219*; Middle-ranked: *r= -.4161, P=.0276*; Subordinate: *r=.1625, P=.4087*; Fig 5d). Correlation analysis between the theta rhythm power of the mPFC and the percentage of correct responses during the *difficult task* showed no significant correlation in all social group of animals (Dominant: *r=.1001, P=.6122*; Middle: *r=.2602, P=.1811*; Subordinate: *r=.01651, P=.9335*; Fig 5e).

**3.2.3. The correlation between theta rhythm power in the mPFC and the latency of correct responses in T-maze with two difficulty levels.** Correlation analysis was performed to examine the correlation of the theta rhythm power of the mPFC and the latency of correct response in the *easy task*. The results revealed significant positive correlation for middle-ranked (Dominant: *r=-.05365, P=.5606*; Middle-ranked: *r=.4514, P<.001*; Subordinate: *r=-.1205, P=.1899*; Fig 5f).

A similar analysis was performed to examine the correlation of theta rhythm power of the mPFC and the latency of correct response during *difficult task*. The results showed that there was a positive trend in subordinate animals (Dominant; *r=.1360, P=.1385*;Middle-ranked: *r=-.08568,P=.3521*;Subordinate:*.r=.2017, P=.0272* Fig 5g).

**3.2.4. Theta rhythm power of the vHPC during responses in T-maze with two difficulty levels.** In vHPC, we compared the theta power of the vHPC during correct and wrong responses of two degrees of task difficulty. Examples of the time-frequency spectrograms and raw LFP of vHPC in the subordinate group are shown in both degrees of task difficulty for correct and wrong responses (Fig 6 a b).

In *easy task*, two-way ANOVA was used to compare the theta power of the vHPC in the correct and wrong responses within each rank and between three different ranks. When we compared *correct and wrong responses* for each social rank, no significant main effect of group was found (F (2, 162) = 1.536, *P=.2184*). Moreover, the response type main effect revealed significant differences between groups (F (1, 162) = 16.09, *P<.001*). However the interaction effect was found no significant (F (2, 162) =.07464, *P=.9281*; Fig 7a).

For *difficult level of task*, we found no significant main effect of group (F (2, 162) =.9193, *P=.4009*; Fig 7b). The main effect of response type showed significant difference between *wrong and correct* responses (F (1, 162) = 88.58, *P<. 001*; Fig 7b). Additionaly, interaction effect was significant (F (2, 162) = 8.438, *P<.001;* Fig 7b). Post-hoc Bonferroni's multiple comparisons test for *correct* responses showed a significant difference between groups; dominant vs. middle-ranked (t (162) = 3.379, *P =.0137*; Fig 7b).

Morover, Bonferroni's multiple comparisons test between correct vs. wrong response revealed significant difference in dominant (t(162)= 3.822, *P =.0028*), subordinate (t(162)= 3.693, *P=.0045*), and the middle (t (162)= 8.787, *P<.001*; Fig 7b) rats.

Using two-way ANOVA to compare the theta power of vHPC in correct responses between *two levels of task difficulty* for three social rank, we found significant main effect of task difficulty (F (1, 162) = 12.82, *P <.001*). Also, significant interaction effect was found (F (2, 162) = 4.936, *P =.0083*). Bonferroni's multiple comparisons test revealed significant difference only in middle-ranked animals (t (162) = 4.618, *P <.001*; Fig 7c).

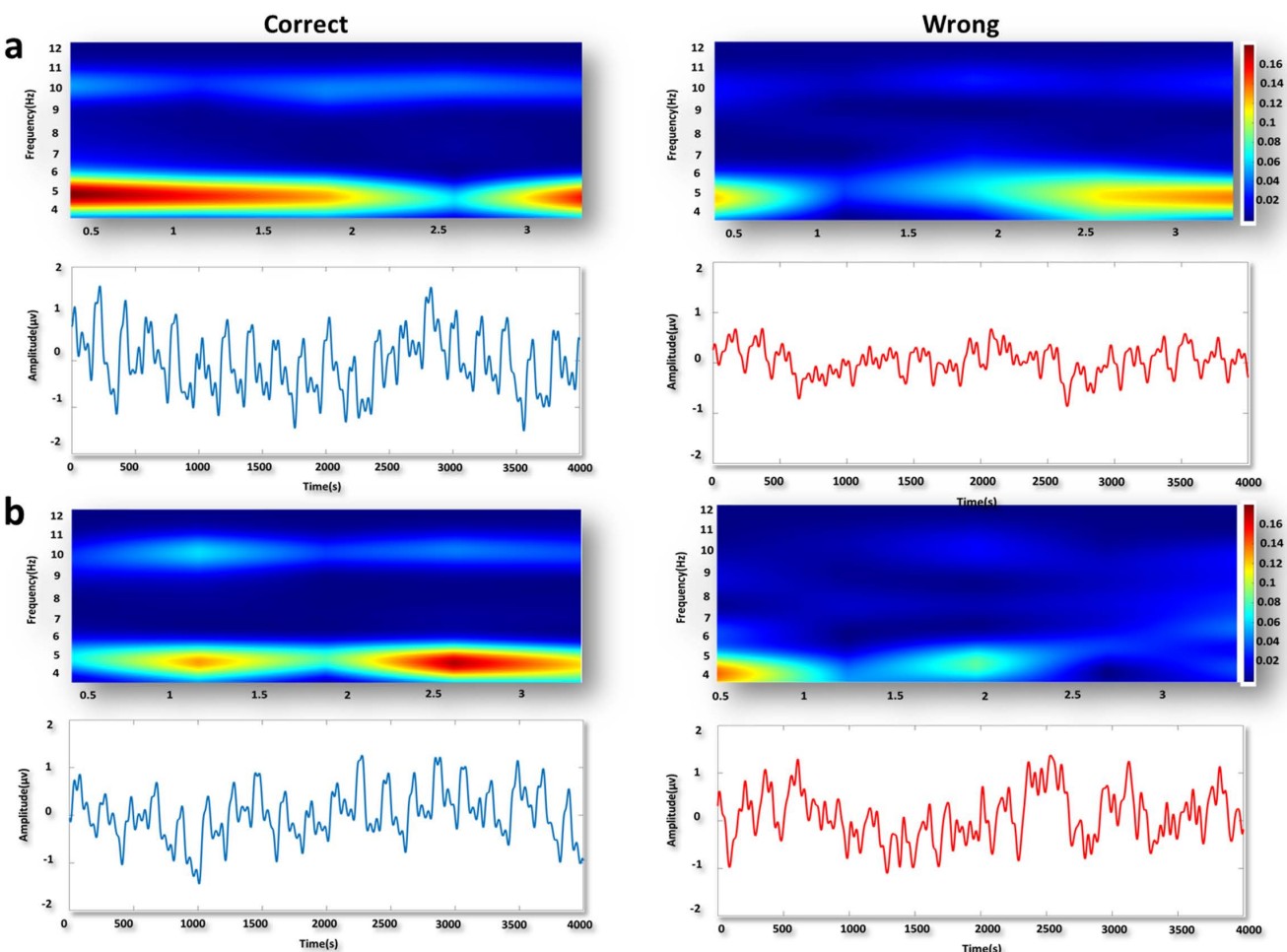

**Fig 6. Sample time-frequency spectrogram and raw LFP signals in vHPC in subordinates.** a) Sample time-frequency spectrogram of one correct trial and its raw signal (blue) and the wrong one (red) into vHPC in easy task (delay: 30s). b) Sample time-frequency spectrogram of one correct trial and its raw signal (blue) and the wrong one (red) in vHPC in difficult task (delay: 5 min).

**3.2.5. The correlation between theta rhythm power in the vHPC and the percentage of correct responses in T-maze with two difficulty levels.** Correlation analysis between the theta rhythm power of the vHPC and the percentage of correct responses during *easy task* revealed posirive significant correlations for subordinate group (Dominant:$r$= 0.08858, $P$=.6540; Middle-ranked: $r$=.1070, $P$=.5878; Subordinate: $r$= 0.4505,$P$=.0161; Fig 7d). Moreover, as the task became more *difficult*, again no significant correlation were found between theta power of the vHPC and the percentage of correct responses for all social ranks (Dominant: $r$=.006783, $P$=.9727; Middle: $r$=-.04357, $P$=.8257; Subordinate: $r$=.2276,$P$=.2442; Fig 7e).

**3.2.6. The correlation between theta rhythm power in the vHPC and the latency of correct responses in T-maze with two difficulty levels.** A correlation analysis was performed to investigate the correlation of theta rhythm power in the vHPC and the latency of correct response in the *easy task*. The results showed significant positive correlations for dominant and subordinate animals (Dominant: $r$=.1950, $P$=.0328; Middle-ranked: $r$=.03183, $P$=.7300; Subordinate:.4021, $P$ <.001; Fig 7f).

In the *difficult task*, a correlation analysis showed a significant positive correlation between theta rhythm power in the vHPC and the latency of correct response in dominant and middle-ranked rats (Dominant: *r*=.2513, *P*=.0056; Middle-ranked: *r*=.1985, *P*=.0297; Subordinate: *r*=.03019, *P*=.7434; Fig 7g).

**3.2.7. Theta range cross-correlation of the mPFC-vHPC circuit during responses in T-maze with two difficulty levels.** In *easy task*, using two-way ANOVA to compare the theta correlation of the mPFC-vHPC circuit between *correct* and *wrong* responses for three social ranks showed significant group main effect (F (2, 162) = 6.106, *P*=.0028), response type main effect (F (1, 162) = 25.16, *P*<.001) and interaction effect (F (2, 162) = 5.634, *P*=.0043; Fig 8a). Bonferroni's multiple comparison tests showed significant differences in *correct* responses between subordinate and dominant (t (162) = 4.570, *P* <.001) and subordinate and

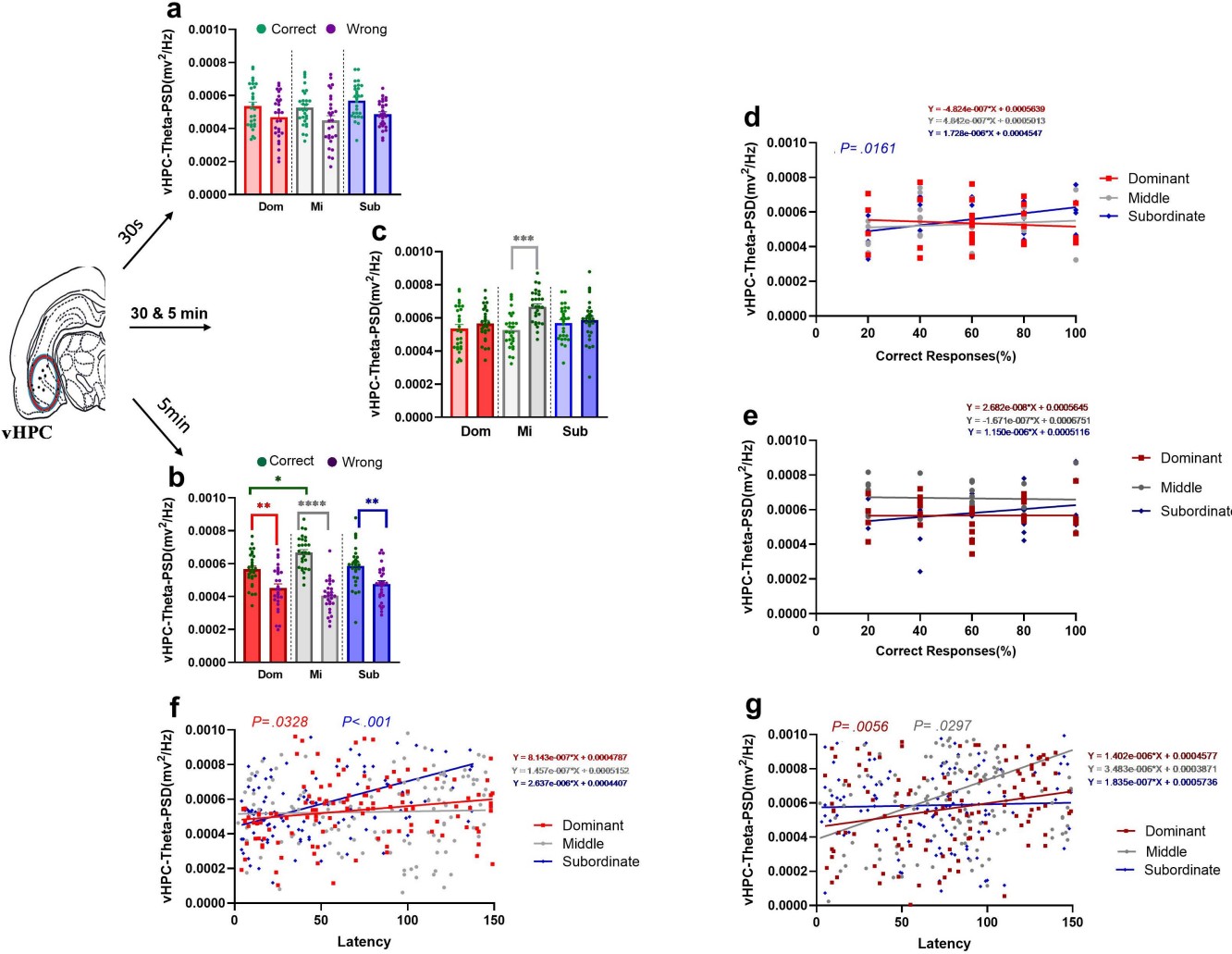

**Fig 7. Mean theta rhythm power (PSDs) of vHPC in T-maze task with two difficulty levels in dominant, middle-ranked, and subordinate animals.** The mean PSDs of vHPC in correct and wrong responses during (a) easy and (b) difficult tasks (* p<.05,** p<.01,**** p<.0001; two-way ANOVA). (c) Comparing PSDs of two degrees of task difficulty in correct responses (***p<0.0001; two-way ANOVA). Values are expressed as mean ±SEM. Correlation of vHPC theta power with the percentage of correct responses during (d) *easy* (S:r=.4505)and (e) *difficult* tasks of three social ranks for 4 days. Correlation of vHPC theta rhythm power with latency of response during (f) *easy* (D: r=.1950, S: r=.4021)and (g) *difficult* (D: r=.2513, M: r=.1985) tasks in three social ranks for 4 days. vHPC: ventral hippocampus, PSD: power spectrum density.

middle-ranked (t (162) = 3.544, $P$=.0077) animals. In respone type comparing Bonferroni's multiple comparisons tests revealed significant difference only in subordinate group(t (162) = 5.460, $P$ <.001; Fig 8a).These results showed subordinate rats had higher theta correlation of the mPFC-vHPC circuit rather than other groups.

In *difficult task*, two-way ANOVA revealed significant difference between theta correlation of the mPFC-vHPC circuit between *correct* and *wrong* responses for three social ranks.There

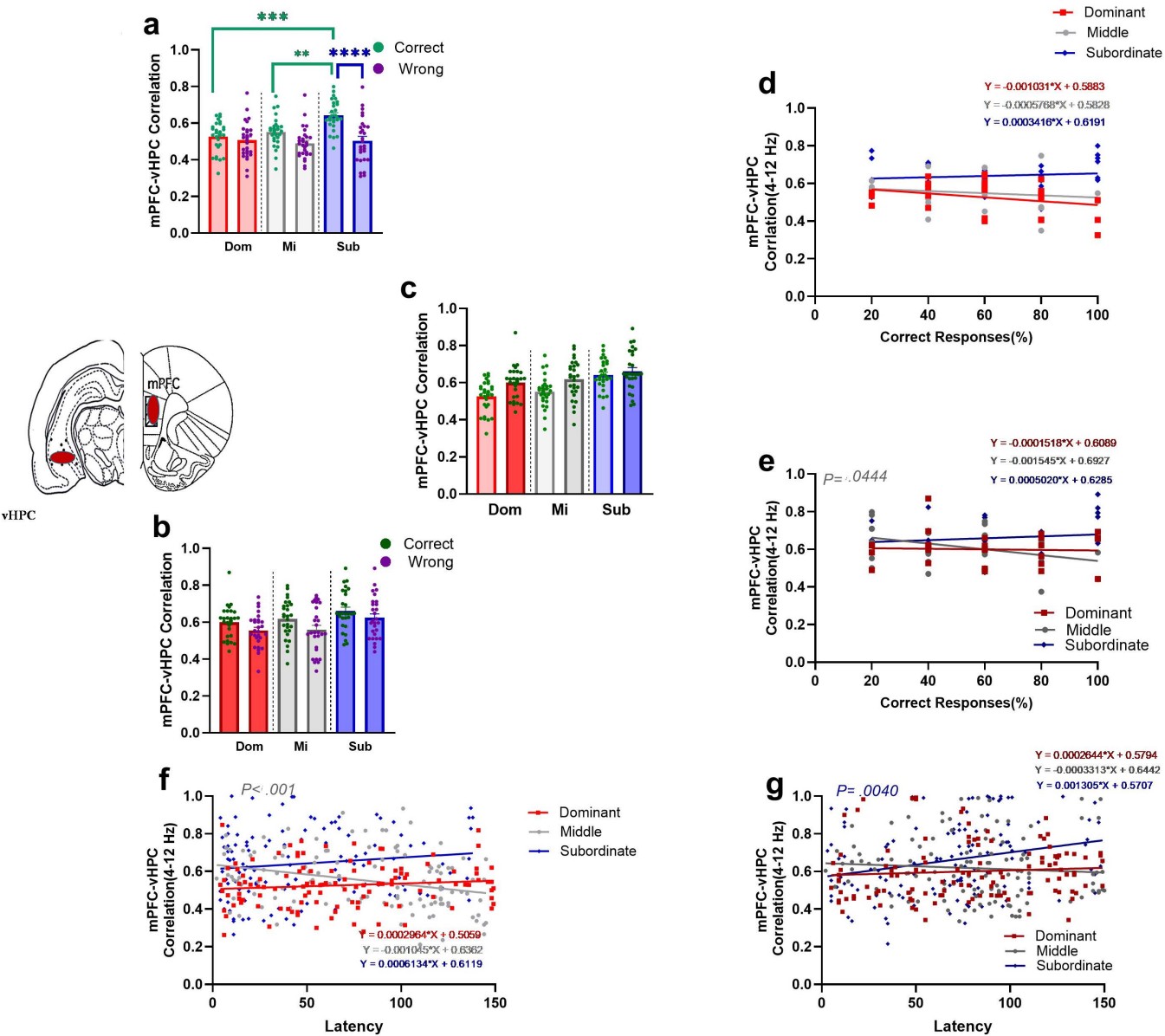

**Fig 8. Mean theta correlation of mPFC-vHPC circuit in T-maze task with two difficulty levels in dominant, middle-ranked, and subordinate animals.** The mean theta correlation of mPFC-vHPC circuit in correct and wrong responses during (a) easy (** p<.001,*** p<.001, **** p<.0001) and (b) difficult tasks. **(c)** Comparing mean theta correlation of two task difficulty in correct responses. Values are expressed as mean ±SEM. Relationship between the theta correlation of mPFC-vHPC circuit with the percentage of correct responses during (d) *easy* and (e) *difficult* (M: r= -.3827)tasks of three social ranks for 4 days. Relationship between the theta correlation of mPFC-vHPC circuit with the latency of correct responses during (f) *easy* (M; r= -.3055)and (g) *difficult* (S:r =.2608) tasks in three social ranks for 4 days.

were significant group main effect (F (2, 162) = 6.014, *P=.0030*) and response type (F (1, 162) = 8.271, *P=.0046*). However, interaction effect was not significant (F (2, 162) =.1451, *P =.8650*; Fig 8b).

Also, we compared the theta range correlation of the mPFC-vHPC circuit in *correct* responses between *two degrees of task difficulty* using two-way ANOVA for each social rank rats. We found significant task difficulty effect (F (1, 162) = 13.67, *P<.001*). However, there was no significant interaction effect (F (2, 162) = 1.396, *P=.2507*).

**3.2.8. Relationship between theta correlation of the mPFC-vHPC circuit and the percentage of correct responses in T-maze with two difficulty levels.** Statistical correlation analysis between the theta range correlation of the mPFC-vHPC circuit and the percentage of correct responses during the *easy task* showed that with the reduction of theta range correlation of the mPFC-vHPC circuit, the percentage of correct responses was stable in all subordinate group (Dominant: *r=-.2936*, *P=.1294*; Middle: *r=-.1701*, *P=.3868*; Subordinate: *r=.1146*, *P=.5615*; Fig 8d).

In the *difficult task*, the relationship between the theta correlation of the mPFC-vHPC circuit and the percentage of correct responses was significant only in middle-ranked (Dominant: *r=-.04396*, *P=.8242*; Middle-ranked: *r=-.3827*, *P=.0444*; Subordinate: *r=.1177*, *P=.5510*; Fig 8e).

**3.2.9. Relationship between theta correlation of the mPFC-vHPC circuit and the latency of correct responses in T-maze with two difficulty levels.** A correlation analysis was performed to investigate the correlation of theta rhythm in mPFC-vHPC circuit and the latency of responses in the *easy task*. The results showed that an increased latency during the easy degree of the task correlated with the decrease of theta rhythm of the two regions in the middle-ranked group (i.e., negative correlation) while the trend has been stable in the dominant and subordinate animals (Dominant: *r=.1104*, *P=.2299*; Middle-ranked: *r= -.3055*, *P<.001*; Subordinate: *r=.1370,P=.1357*; Fig 8f).

In addition, correlation analysis was performed to investigate the correlation between the theta rhythm of two regions and the latency during the *difficult task*. The results showed that with an increase in the latency of responses, the correlation of theta rhythm between mPFC and vHPC increased (i.e., positive correlation) only in subordinates (Dominant: *r=.07582*, *P=.4105*; Middle-ranked: *r= -.07403*, *P=.4216*; Subordinate: *r=.2608*, *P=.0040*; Fig 8g).

**3.2.10. Theta range coherency of the mPFC-vHPC circuit during responses in T-maze with two difficulty level.** We compared the theta coherency of the mPFC-vHPC circuit in the correct and wrong responses using two-way ANOVA test. In the *easy task*, when we compared *correct and wrong responses* for each social rank, it was found significant differences. The group main effect was significant (F (2, 162) = 81.01,*P<.001*). Additionally, there was significant response type main effect (F (1, 162) = 1529, *P<.001*) and interaction effect (F (2, 162) = 65.43, *P<.001*; Fig 9a).

Moreover, Bonferroni's multiple comparisons tests revealed significant differences in *correct* responses for subordinate vs. dominant (t (162) = 9.438, *P <.001*), subordinate vs. middle-ranked (t (162) = 17.06, *P <.001*), and dominant vs. middle-ranked (t (162) = 7.619, *P<.001*) animals. In *wrong* responses, Bonferroni's multiple comparisons tests revealed no significant differences between groups. Bonferroni's multiple comparisons tests revealed significant differences between correct and wrong responses in the dominant (t (162) = 22.07, *P <.001*); the middle-ranked (t (162) = 14.75, *P <.001*), and subordinate (t (162) = 30.90, *P <.001*; Fig 9a) rats. According to these results subordinate rats indicated higher theta coherency of the mPFC-vHPC circuit.

In the *difficult task*, comparing the coherency of the circuit between *correct* and *wrong* responses for three social ranks using two-way ANOVA, there were significant group main

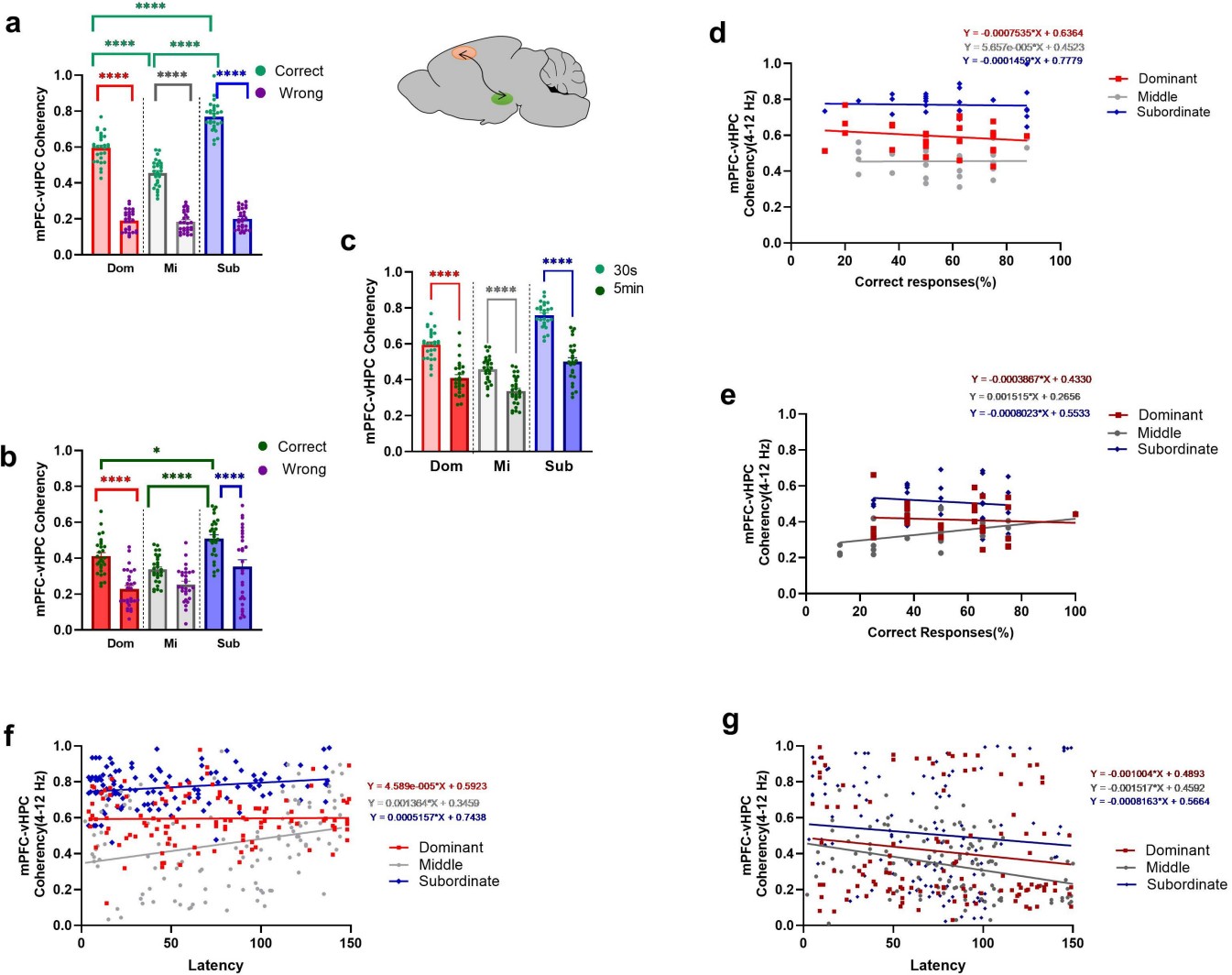

**Fig 9. Mean theta coherency of mPFC-vHPC circuit in T-maze task with two difficulty levels in dominant, middle-ranked, and subordinate animals.** The mean theta coherency of the mPFC-vHPC circuit in correct and wrong responses during (a) *easy* (**** p<0.0001) and (b) *difficult* (* p<0.05, **** p<0.0001) tasks. **(c)** Comparing mean theta coherency of two degrees of task difficulty in correct responses (**** P<0.0001). Values are expressed as mean ±SEM. Correlation between theta coherency of mPFC-vHPC circuit with the percentage of correct responses during (d) *easy* (S: r=.4329) and (e) *difficult* (D: r=.5030) tasks of three social ranks for 4 days. Correlation between theta coherency of mPFC-vHPC circuit with the latency of correct responses during (f) *easy* (M: **r** =.2682, S: r=.1991)and (g) *difficult* (M: r= -.3471)tasks in three social ranks for 4 days.

effect (F (2, 162) = 20.09, *P<.001*), response type (F (1, 162) = 56.64, *P<.001*). However, no significant interaction effect was revealed (F (2, 162) = 2.539, *P=.0820*, Fig 9b). Bonferroni's multiple comparisons tests revealed significant differences in *correct* responses for subordinate vs. middle-ranked (t (162) = 5.320, *P <.001*), subordinate vs. dominant (t (162) = 3.024, *P =.0435*) animals. Additionally there were significant differences between correct and wrong reponses in dominant (t (162) = 5.658, *P <.001*), and subordinate (t (162) = 4.805, *P <.001*; Fig 9B).

Finally, using two-way ANOVA test, we compared coherency of mPFC-vHPC circuit in *correct* responses between *two levels of task difficulty* for among social ranks. The result showed a significant difference in main effect of task difficulty (F (1, 162) = 187.6, *P<.001),* and interaction effect (F (2, 162) = 8.923, *P <.001;* Fig 9c). Bonferroni's multiple comparisons

test revealed significant difference in dominant (t (162) = 7.769, *P <.001*), middle (t (162) = 4.992, *P <.001*), and subordinate (t (162) = 10.96, *P <.001*) animals.

**3.2.11. The relationship between theta range coherency of the mPFC-vHPC circuit and the percentage of correct responses in T-maze with two difficulty levels.** Examining the relationship between theta coherency of the mPFC-vHPC circuit and the percentage of correct responses in the *easy task* showed positive significant correlations in subordinate group (Dominant: *r*= -.1866, *P*=.3416; Middle-ranked: *r*=.1527, *P*=.4378; Subordinate: *r*=.4329, *P*=.0214; Fig 9d).

In the *difficult task*, the theta coherency of the mPFC-vHPC circuit was positively correlated with the percentage of correct responses in subordinate group (Dominant: *r*=.1328, *P*=.5006; Middle-ranked: *r*= -.1643, *P*=.4035; Subordinate: *r*=.5030, *P*=.0064; Fig 9e).

**3.2.12. The relationship between coherency of the mPFC-vHPC circuit and the latency of correct responses in T-maze with two difficulty level.** A correlation analysis was performed to investigate the relationship between theta range coherency in the mPFC-vHPC circuit and the latency of *correct* responses in the *easy task*. The results showed that with the increased time to reach the target arm, theta range coherency between these two areas was increased (i.e., positive correlation) in all social groups (Dominant: *r*=.01415, *P*=.8781; Middle-ranked: *r*=.2682, *P*=.0031; Subordinate: *r*=.1991, *P*=.0293; Fig 9f).

In the *difficult task*, the correlation between coherency of these two areas and the latency to reach the target arm was negative in all social groups(Dominant: *r*=-.1448, *P*=.1145; Middle: *r*=-.3471, *P<. 001*; Subordinate: *r*=-.1056, *P=.2511*; Fig 9g).

## 4. Discussion

This study investigated the impact of social hierarchy on SWM performance in different social statuses of male rats, focusing on the mPFC-vHPC circuit for the first time, a common neural circuit implicated in both social hierarchy and SWM. In this study, our behavioral findings indicated that subordinate rats had better performance in easy degrees of task difficulty compared to other ranks. Moreover, electrophysiological analysis of the mPFC-vHPC circuit revealed higher correlation and coherency in the mPFC-vHPC circuit of the subordinate group in the easy task and higher coherency in difficult task.

In behavioral tests, the subordinate group indicated shorter latency in behavioral sessions of the *easy task, but not in difficult task.* These findings indicate better memory performance in subordinates, which appears to be in contrast with general view regarding poor cognitive function in subordinate rats. This highlighs the importance of controlling many confounding factors that might influence the result of cognitive performance, such as genetic background, access to favorable resources, mating experiences, and post-hierarchy living conditions [28–32] that were mostly controlled in our study. A recent study by Jaafari et al. (2022) which is in line with our findings indicated that SWM performance was lower in subordinates compared to dominants before formation of dominance hierarchy but it was reversed after hierarchy formation [10]. This aligns with our behavioral results in easy task that subordinate rats performed better in SWM task compared to higher ranked animals. Subordinate group showed more non-significant correct responses in both degrees of task difficulty.

Our electrophysiological study showed that the power of theta rhythm in the mPFC and vHPC was similar in dominant and subordinate animals during easy and difficult task performance. For middle-ranked animals, the theta power of mPFC was lower during easy task that were matching with the results of SWM. Moreover, higher correlation and coherency of theta power between mPFC and vHPC during easy task were matching with behavioral performance of subordinate rats in T-maze. According to the results in the easy task, the theta

rhythm power in the mPFC in the subordinate group positively correlated with the percentage of correct responses and in vHPC. In the latency, the theta rhythm power in the mPFC negatively correlated with task latency in subordinates but not in the vHPC. Theta correlation of the mPFC-vHPC circuit in the subordinate group is stable with the percentage of correct responses and latency during the easy degree of task.Theta range coherency of the mPFC-vHPC circuit is positively correlated with the percentage of correct responses during the easy degree of task in subordinate group. Also, the theta range coherency of the mPFC-vHPC circuit positively correlated with latency in the easy degrees of the task. When the task gets difficult, the theta rhythm power in the mPFC in the subordinate group positively correlated with the percentage of correct responses and latency. In vHPC, there is no correlation between the percentage of correct responses and latency in subordinate group. Theta correlation of the mPFC-vHPC circuit was correlated positively with the percentage of correct responses and latency in subordinate group. Theta range coherency of the mPFC-vHPC circuit was positively correlated with the percentage of correct responses but negatively with the latency during the difficult degree of the task in the subordinate group.

Several decades of studies have highlighted inevitable top-down controling role of the PFC in memory processing. Studies have shown that intact PFC is essential for learning and substitution of memory strategies and updating information in various memory tasks [13]. The considerable feature of mPFC neurons is their sustained activity during performing memory tasks that require the maintenance of information in a given time slot [33,34]. In previous studies, it was found that the theta waves of the mPFC region showed greater neural activity and higher correlation of the theta phase of this area with the surrounding neurons in subordinate rats during spatial exploration [23]. Lower theta power of mPFC in middle-ranked animals during easy task corresponds with higher latency in decision point compared to subordinates.

In the other hand, previous studies indicated that the expression of hippocampal genes (e.g., Phf2, Creb, Grin2b, GluR1) differed in dominant and subordinate rats [30]. Also, it was found that the sharp wave ripples (SWRs) in the HPC were significantly associated with social rank. The SWRs amplitude in dominant rats was larger as compared to the subordinate rats and the SWRs frequency of the subordinate rats was slower [22]. The vHPC is necessary for recalling episodic memories. In this way, this pattern is determined in the firing of hippocampal neurons to represent a special object or events in a specific place or time. Our results showed more non-significant vHPC power in the subordinate group rather than other groups and, it means subordinate group tried more to coding environmental cues.

So, the HPC is responsible for registering new information, while the PFC is responsible for flexible changes based on the changes in the rules of the context, especially the rules that the HPC has registered [35]. In this way, the results can be interpreted that dominant and subordinate rats show more theta power in the mPFC region and have more neuronal activity in this region.

In the *difficult* task, no differences were observed in mPFC theta power among three ranks. This suggests that task difficulty may affect mPFC activity uniformly across social hierarchy levels. In general, there are limited studies based on task difficulty in SWM, and the majority of these studies have been behavioral investigations. It was found that memory performance decreases with increasing task difficulty [36]. In another study, the function of mPFC was investigated by increasing the difficulty of the task during the execution of a memory task. It was found that as the delay time in the task increased and became more difficult, damage to the mPFC had less determining role on memory performance [25]. The mPFC region is not involved in coding of information during longer delays but is responsible for managing necessary processes to adapt the information to new possibilities for planning decisions. So, the equal mean power in mPFC at this level of difficulty indicates that three

social groups had similar engagement of this region in making a flexible decision to change the rules [34]. About vHPC, as the difficulty of the task was increased, a significant increase was shown in the - middle-ranked animals compared to other ranks while no difference was found in the theta power between dominant and subordinate groups. The increased activity of the vHPC in middle-ranked rats during difficult task seems to be related to increased effort to respond quickly at the decision point though less successful compared to subordinates [8,37].

In the next step of electrophysiological investigations, the theta correlation of the mPFC-vHPC circuit and the coherence of the theta oscillations of this circuit were examined. In the easy degree of task difficulty, the subordinate group showed the highest correlation in line with the behavioral results compared to other two higher ranks. This means that the mPFC and vHPC regions showed more similarity and synchronicity in the power of their waves in the subordinate group that resulted in a faster response [38]. Based on the studies, the correlation of theta waves shows a significant increase when animals are actively searching the environment to respond to memory tasks [34,37]. In the network cooperation between these two areas, it has been determined that the activity of these two areas increases simultaneously until they reach the decision point, but after passing this point, this correlation decreases [8]. Also, studies indicated that the increased bidirectional information flow in the vHPC-mPFC network during the SWM task and the increase in information flow was related with memory accuracy. So this correlation is associated with efficient behavioral performance [23]. Theta correlation of these two areas in the difficult degree of task was the same between groups. It may implies that as the task became more difficult, the animals' effort to complete the task approaches maximal level [23,38].

Considering coherency in the easy task, the subordinate group displayed the highest coherence in the mPFC-vHPC circuit, corresponding with better SWM performance. Liu et al (2018) showed theta oscillations in the mPFC-vHPC circuit are important for information flow that appear to underlie network-wide information storage. In this way, the higher coherence of this circuit predicts better performance in the memory task and reducing this coherence will decrease the performance to the random level [8]. Bidirectional information flow between these two regions shows greater coherence during correct responses than during wrong responses and thus predicts response accuracy [8]. Therefore, based on the behavioral results related to response accuracy, the subordinate rats had relatively faster performance for the same level of accuracy compared to high-ranked rats with higher correlation and coherency at the level of neural circuits.

In another study, resarchers investigated the role of mPFC and vHPC at different levels of working memory task difficulty, they found that bilateral and simultaneous inactivation of the mPFC and vHPC regions causes more wrong answers in difficult task (5 min), while in the easy task (10 s), their performance in SWM was not affected much. Accordingly, when the task becomes more difficult due to the increase in cognitive load to complete the task, the simultaneous activity of both areas becomes necessary, while in the case of an easier task, each of the areas alone can process separately and guide the completion of the task [25]. The reason for this may be explained based on the different roles of HPC (acquiring, storing, and retrieving information) and mPFC (altering and replacing information) during the execution of the SWM task. Since in long delays, a change in memory strategy is evident, the involvement of mPFC is necessary to organize information. On the other hand, as the HPC is responsible for encoding information, the simultaneous activity of both area as a functional unit is necessary to perform the task with longer delay [25].

Recently, it has been shown dominant rats had significantly higher anxiety-like behaviors compared to subordinates as well as the spine density analysis revealed a significantly

higher number of spines in subordinates compared to the dominant rats in dmPFC pyramidal neurons and the apical and basal dendrites of hippocampal CA1 pyramidal neurons [39]. So, one potential explanation for the differences in cognitive and neural activity between dominants and subordinates could be structural variations in key brain regions that determines the anxiety levels among rats across different social ranks. Given that the mPFC and vHPC regions are simultaneously active during anxious behaviors, and considering that theta power in the mPFC region increases during such behaviors, differences observed in this study in social hierarchies may stem from varying anxiety levels among rats. Considering the effect of tube test on anxiety-like behaviors, previous research has shown that the anxiety levels of dominant and subordinate rats remain consistent before and after the tube test, suggesting minimal impact on behavioral task performance [40].

Comparing two degrees of task difficulty, it was found that when the task gets difficult, the mPFC-vHPC circuit activity is associated with a decrease in the coherency of theta rhythm between these two areas. Reduced mPFC-vHPC theta coherency during difficult task for all social ranks rats could be a sign of reduced caring toward the target selection. Future studies using some other SWM tests may shed more light on the relationship between engaged neural circuits and cognitive manifestations of social hierarchy. Using extensive cognitive tests to confirm these results is strongly recommended for the future studies regarding social rank and cognitive performance. The spontaneous alternations in T-maze without training or motivation for favorable resources may have created some limitations regarding cognitive challenges of the difficult task with longer delay. The relationship between dopamine efflux in the mPFC and foraging for food based on working memory is also discussed in support of the conjecture that dopamine may serve as a link between motivation and memory functions. It is probable that less motivation to choose correct arm during difficult task has resulted in somehow random selection of the arms and non-significant difference between different social ranks [41]. According to our findings, in this study, the subordinate rats showed better performance at the behavioral and electrophysiological levels compared to the other two groups. At the behavioral level, the subordinate rats were able to respond faster in choosing the target arm, as well as have a higher response accuracy, and at the level of electrophysiology, they had a higher theta correlation and coherence of mPFC-vHPC than the other two ranks. As the task becomes more difficult, the behavioral performance of the dominant rats remains unchanged, while the subordinate rats experience a decrease in performance. At the level of electrophysiology, in all three social groups, the coherence of the circuit decreases with the increases in the difficulty of the task. This shows that the dominant rats were able to maintain the same behavioral performance despite reduced mPFC-vHPC coherency.

## 5. Conclusion

In summary, this study revealed that subordinate rats performed better than higher ranked animals in short-delay SWM task in T-maze spontaneous alternations. LFP recordings from vHPC and mPFC revealed higher theta rhythm power correlation and coherency between these two brain regions in subordinates during decision point of easy task in T-maze. It seems that increasing delay period and difficulty of the task in spontaneous alternation without motivation for favorable resources has diminished caring about target arm selection as is evident by reduced coherency between mPFC and vHPC. Future studies using working memory tasks that maintains higher motivation during longer-delay tasks may be of choice to determine the relationship between social hierarchy, cognitive function, and the activity of the underlying neural circuits.

## Author contributions

**Conceptualization:** Soomaayeh Heysieattalab, Ali Jaafari suha.

**Data curation:** Farhad Farkhondeh Tale Navi.

**Formal analysis:** Faezeh Zarfsaz, Hamid Basiryan.

**Investigation:** Faezeh Zarfsaz.

**Methodology:** Ali Jaafari suha.

**Supervision:** Soomaayeh Heysieattalab.

**Validation:** Soomaayeh Heysieattalab, Farhad Farkhondeh Tale Navi.

**Visualization:** Farhad Farkhondeh Tale Navi.

**Writing – original draft:** Faezeh Zarfsaz.

**Writing – review & editing:** Soomaayeh Heysieattalab, Ali Jaafari suha, Farhad Farkhondeh Tale Navi.

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
