## [Decision Letter · Decision Letter 0]

29 Sep 2024

PONE-D-24-36279Social subordination is associated with better cognitive performance and higher theta coherence of the mPFC-vHPC circuit in male ratsPLOS ONE

Dear Dr. Heysieattalab,

Thank you for submitting your manuscript to PLOS ONE. After careful consideration, we feel that it has merit but does not fully meet PLOS ONE’s publication criteria as it currently stands. Therefore, we invite you to submit a revised version of the manuscript that addresses the points raised during the review process.

**ACADEMIC EDITOR:** - please do revise the manuscript significantly in order to address adequately all the comments of the reviewers. Do make manuscript more concise and emphasize the novel findings.  

We look forward to receiving your revised manuscript.

Kind regards,

Dragan Hrncic

Academic Editor

PLOS ONE

Journal Requirements:

1. When submitting your revision, we need you to address these additional requirements. Please ensure that your manuscript meets PLOS ONE's style requirements, including those for file naming. The PLOS ONE style templates can be found at https://journals.plos.org/plosone/s/file?id=wjVg/PLOSOne_formatting_sample_main_body.pdf and https://journals.plos.org/plosone/s/file?id=ba62/PLOSOne_formatting_sample_title_authors_affiliations.pdf 2. To comply with PLOS ONE submissions requirements, in your Methods section, please provide additional information regarding the experiments involving animals and ensure you have included details on (1) methods of sacrifice, (2) methods of anesthesia and/or analgesia, and (3) efforts to alleviate suffering. 3. We note that the grant information you provided in the ‘Funding Information’ and ‘Financial Disclosure’ sections do not match.  When you resubmit, please ensure that you provide the correct grant numbers for the awards you received for your study in the ‘Funding Information’ section. 4. Thank you for stating the following financial disclosure: "We would thank the Iran National Science Foundation for foundation support (grant number: 4012752)." Please state what role the funders took in the study.  If the funders had no role, please state: "The funders had no role in study design, data collection and analysis, decision to publish, or preparation of the manuscript." If this statement is not correct you must amend it as needed. Please include this amended Role of Funder statement in your cover letter; we will change the online submission form on your behalf. 5. Thank you for stating the following in the Acknowledgments Section of your manuscript: "We would thank the Iran National Science Foundation for foundation support (grant number: 4012752)". We note that you have provided funding information that is not currently declared in your Funding Statement. However, funding information should not appear in the Acknowledgments section or other areas of your manuscript. We will only publish funding information present in the Funding Statement section of the online submission form. Please remove any funding-related text from the manuscript and let us know how you would like to update your Funding Statement. Currently, your Funding Statement reads as follows: "We would thank the Iran National Science Foundation for foundation support (grant number: 4012752)." Please include your amended statements within your cover letter; we will change the online submission form on your behalf. 6. In the online submission form, you indicated that your data is available only on request from a third party. Please note that your Data Availability Statement is currently missing the contact details for the third party, such as an email address or a link to where data requests can be made. Please update your statement with the missing information. 

Reviewers' comments:

Reviewer's Responses to Questions

**Comments to the Author**

1. Is the manuscript technically sound, and do the data support the conclusions?

Reviewer #1: Partly

Reviewer #2: Yes

2. Has the statistical analysis been performed appropriately and rigorously? 

Reviewer #1: I Don't Know

Reviewer #2: Yes

3. Have the authors made all data underlying the findings in their manuscript fully available?

Reviewer #1: Yes

Reviewer #2: Yes

4. Is the manuscript presented in an intelligible fashion and written in standard English?

Reviewer #1: Yes

Reviewer #2: Yes

5. Review Comments to the Author

Reviewer #1: The authors investigated the relationship between social dominance hierarchies and neural connections between the medial prefrontal cortex (mPFC) and the ventral hippocampus (vHPC). The authors then addressed the role of social hierarchy and spatial working memory (SWM). Although it is a very interesting research topic, the vast amount of data and lengthy papers often confuse readers. This paper contains a lot of information that is already known, and it is important to cite previous research and note any discrepancies between the authors' results and those of other studies. What were the new findings in the study conducted by the authors? Also, what has already been clarified? Please organize these. We would like you to revise your paper with consideration to summarizing it concisely.

For example, it is important to keep the abstract focused on the data that address new findings from the study and to summarize explanations of non-significant data concisely, limiting interpretations to those that are specifically needed to support the discussion. In addition, in terms of materials and methods, the main conditions should be those that allow the experiment to be reproduced with good reproducibility, and it is appropriate to include the numerical values etc. required for the experiment. The T-maze is a widely used method in behavioral analysis, so explanations of it will be kept to a minimum; more important are details about lighting, time of measurement, etc. General methods should be described by simply saying "according to standard methods," and if the authors' original methods or measurement conditions are special, these should be described. The results section is often filled with data stating that no significant differences were observed. If data is particularly necessary, it is necessary to include it, but if it is not, and can be understood by looking at figures, etc., it may be possible to omit it. Although it is a minor point, the way of writing p-values should be standardized (p<0.001 or p<.001). Also, significant figures should be considered. Regarding the writing style of papers, figure legends are currently written in the main text of papers, but it is common to put them together at the end of the main text.

Overall, it is not clear in reviewing this study what is novel. The experiments in this study included pilot and follow-up experiments, and ideally the main data should be those experiments that focused on the purpose of the study. The paper should cite references to what has already been published in previous research papers, mention what the authors have newly clarified in this study, and focus on the methodology used to arrive at this conclusion and the data necessary for discussion.

Reviewer #2: The ms by Zarfsaz and colleagues describes the oscillatory activity in the mPFC-vHPC circuit (mainly exploring correlation and coherence) in both social hierarchy and spatial working memory.

Behavioral analysis revealed that subordinate rats had better performance in easy degrees of task difficulty compared to others, while neurophysiological data indicated in the easy task a higher correlation and coherency in the mPFC-vHPC circuit in the subordinate group and only higher coherency in difficult task.

The ms is interesting, though some points need to be clarified

1. Given the dHPC's primary role in spatial working memory, alongside the vHPC, the authors should provide a more detailed explanation for their decision to focus solely on the ventral part of this brain region.

2. A more detailed explanation of the system used to determine the precise timing of the decision point is necessary. Did they use video-EEG?

3. I have some questions about the T-maze task. Could the authors please provide a more detailed description of the protocol? Did they impose a time limit for the animals to complete the task? Were any animals excluded from the behavioral testing?"

Given that mice exhibited decreased speed and increased latency over successive trials, did the authors observe any variations in performance within a single day or across the eight-day testing period?

Do the T-maze results represent the cumulative number of correct and incorrect responses across the eight days of trials? This information should be clarified in the Methods or Figure legend. Additionally, were LFP analyses conducted for each individual trial?

4. Regarding Figure 1, authors should improve the pictures displaying the brain section in which the electrodes were implanted and describe how did they acquired the image.

5. In the abstract I presume that the sentence "animals were cannulated" refers to electrodes implant; rephrase.

6. PLOS authors have the option to publish the peer review history of their article (what does this mean?). If published, this will include your full peer review and any attached files.

Reviewer #1: No

Reviewer #2: No

---

## [Author Response · Author response to Decision Letter 1]

4 Dec 2024

#Journal Requirements:

1.When submitting your revision, we need you to address these additional requirements

Answer: Thanks for your comments, the article has been changed to the requested format.

Answer: Thanks for your comment, all points have been addressed and highlighted in the manuscript.

#method: To implant the electrodes in the specified areas, at first the rat was anesthetized using a mixture of ketamine (100 mg/kg) and xylazine (10 mg/ kg). Then, the anesthetized rat was placed in the stereotaxic apparatus (Toosbioresearch, Mashhad, Iran) and a heating pad was used to maintain the rat body temperature at 37 °C during surgery. Anesthesia depth was checked by tail and foot pinch reflexes. Vitamin A ointment was also used to prevent the animal's eyes from drying out during surgery.

To ensure accurate electrode placement, rats were first deeply anesthetized with carbon dioxide and then the brains of the rats were carefully removed and placed in 4% paraformaldehyde at 4°C for 48h. Then, using a vibroslicer, the brains were cut and stained with methylene blue and examined under a microscope (AC 230V 50 Hz, Fig 1b).

Answer: Thank you for notifying us about this matter, we considered this point during resubmission.

4. Thank you for stating the following financial disclosure: "We would thank the Iran National Science Foundation for foundation support (grant number: 4012752)." Please state what role the funders took in the study. If the funders had no role, please state: "The funders had no role in study design, data collection and analysis, decision to publish, or preparation of the manuscript."

Answer: The funders had no role in study design, data collection and analysis, decision to publish, or preparation of the manuscript.

5. Thank you for stating the following in the Acknowledgments Section of your manuscript: "We would thank the Iran National Science Foundation for foundation support (grant number: 4012752)".

We note that you have provided funding information that is not currently declared in your Funding Statement. However, funding information should not appear in the Acknowledgments section or other areas of your manuscript. We will only publish funding information present in the Funding Statement section of the online submission form. Please remove any funding-related text from the manuscript and let us know how you would like to update your Funding Statement. Currently, your Funding Statement reads as follows: "We would thank the Iran National Science Foundation for foundation support (grant number: 4012752)." Please include your amended statements within your cover letter; we will change the online submission form on your behalf.

Answer: Thanks for mentioning this point, the mentioned part was deleted from the manuscript and was added to cover letter.

6. In the online submission form, you indicated that your data is available only on request from a third party. Please note that your Data Availability Statement is currently missing the contact details for the third party, such as an email address or a link to where data requests can be made. Please update your statement with the missing information.

Answer: We are sorry for the inconvenience. Thank you for reminding us.

Data Availability: The datasets generated during and/or analyzed in the current study are available from the corresponding author upon reasonable request.

#Reviewer1

1. The authors investigated the relationship between social dominance hierarchies and neural connections between the medial prefrontal cortex (mPFC) and the ventral hippocampus (vHPC). The authors then addressed the role of social hierarchy and spatial working memory (SWM). Although it is a very interesting research topic, the vast amount of data and lengthy papers often confuse readers. This paper contains a lot of information that is already known, and it is important to cite previous research and note any discrepancies between the authors' results and those of other studies. What were the new findings in the study conducted by the authors? Also, what has already been clarified? Please organize these. We would like you to revise your paper with consideration to summarizing it concisely. findings from the study and to summarize explanations of non-significant data concisely, limiting interpretations to those that are specifically needed to support the discussion.

Answer: Thank you very much for pointing these out, the text of the manuscript were rearranged and presented in the following manner as was stated by the esteemed reviewer. The changes are highlighted in the revised manuscript text.

#Abstract: Electrophysiological data revealed significant theta correlation and coherence of the mPFC-vHPC circuit that was higher in subordinates. By increasing task difficulty, theta rhythm coherency was reduced for all social ranks but subordinates maintained better task performance with less reduction of theta coherence. These findings are presented for the first time and underscore the association between different social hierarchies and working memory performance within the mPFC-vHPC circuit; highlighting the influence of social rank on implicated circuit.

#Introduction: Considering controversial results from different studies regarding relationship between social hierarchy and memory performance, our study tries for the first time to elucidate the impact of social hierarchy on SWM during two degrees of task difficulty, specifically aiming at shedding light on the underlying functional connectivity within the mPFC-vHPC circuit by LFP recording in sibling male rats with different social ranks.

#Results: In this section, an attempt was made to omit parts of the results that were non- significant and were indicated with track changes in the text.

#Discussion: This study investigated the impact of social hierarchy on SWM performance in different social statuses of male rats, focusing on the mPFC-vHPC circuit for the first time, a common neural circuit implicated in both social hierarchy and SWM.

According to our findings, in this study, the subordinate rats showed better performance at the behavioral and electrophysiological levels compared to the other two groups. At the behavioral level, the subordinate rats were able to respond faster in choosing the target arm, as well as have a higher response accuracy, and at the level of electrophysiology, they had a higher theta correlation and coherence of mPFC-vHPC than the other two ranks. As the task becomes more difficult, the behavioral performance of the dominant rats remains unchanged, while the subordinate rats experience a decrease in performance. At the level of electrophysiology, in all three social groups, the coherence of the circuit decreases with the increases in the difficulty of the task. This shows that the dominant rats were able to maintain the same behavioral performance despite reduced mPFC-vHPC coherency.

2. In addition, in terms of materials and methods, the main conditions should be those that allow the experiment to be reproduced with good reproducibility, and it is appropriate to include the numerical values etc. required for the experiment. The T-maze is a widely used method in behavioral analysis, so explanations of it will be kept to a minimum; more important are details about lighting, time of measurement, etc. General methods should be described by simply saying "according to standard methods," and if the authors' original methods or measurement conditions are special, these should be described.

Answer: Thank you for this point, we have provided more details about situation of conducting T-maze protocol.

#method: The rats were obtained from Urmia Medical Sciences University and housed at 21±2 °C, 12:12 light-dark cycle (from 8 am to 8 pm). They were kept in standard animal research facilities in which food and water were freely available.

According to the standard protocol for T-maze running [23], we used the spontaneous alternation which is performed without a food reward and habituation. Here, it is the novelty of the maze arms that drives the spontaneous exploration. We used this protocol because of rats’ difference in desire for food between dominant and subordinate rats [27]. In T-maze task, for the sample phase, each rat was first placed at the starting point (i.e., the base of the arm perpendicular to the left and right arms) of the maze and was let to choose between left or right arms. After the sample phase, the rat was locked in the selected arm for 30 seconds and then returned to the starting point. At the starting point, depending on the difficulty of the task, rats were locked at that point for 30 s (easy) or 5 min (difficult), a delay time between sample and choice run trials. In this protocol, it is possible to freely choose the target arm in both the sample phase and choice phase [23]. The difficulty of the task was alternated every day for eight days (four days of 30 seconds difficulty and four days of 5 minutes). Six trials were performed every day (one sample trial and five choice trials). The time limit for the rats to remain in the maze was 180 seconds [23]. All movements of the animals were recorded via a video camera located above the maze for subsequent analysis. All tests were conducted in the afternoon every day from 3 pm to 6 pm. Also, the light intensity during the experiment was 100 lux.

3. The results section is often filled with data stating that no significant differences were observed. If data is particularly necessary, it is necessary to include it, but if it is not, and can be understood by looking at figures, etc., it may be possible to omit it. Although it is a minor point, the way of writing p-values should be standardized (p<0.001 or p<.001). Also, significant figures should be considered. Regarding the writing style of papers, figure legends are currently written in the main text of papers, but it is common to put them together at the end of the main text.

Answer: Thanks for your comments, all of the changes have been applied carefully. Some of the non-significant data were particularly necessary, so they remained but some of them weren’t necessary and they have been omitted and indicated with track changes in the manuscript. In the manuscript, all the changes have been highlighted in the Results section. Also, we have written the p-values in standardized form (p<.001). All figure legends are put together at the end of the main text.

4. Overall, it is not clear in reviewing this study what is novel. The experiments in this study included pilot and follow-up experiments, and ideally the main data should be those experiments that focused on the purpose of the study. The paper should cite references to what has already been published in previous research papers, mention what the authors have newly clarified in this study, and focus on the methodology used to arrive at this conclusion and the data necessary for discussion.

Answer: Sorry to cause confusion, according to the answer given to the question one, the changes in introduction are as followed “Considering controversial results from different studies regarding relationship between social hierarchy and memory performance, our study tries for the first time to elucidate the impact of social hierarchy on SWM during two degrees of task difficulty, specifically aiming at shedding light on the underlying functional connectivity within the mPFC-vHPC circuit by LFP recording in sibling male rats with different social ranks. Therefore, according to the literature of research, which were mainly behavioral studies, and in some of them the subordinate group performed better [10] and in others the dominant group [21,22], our study tried for the first time to compare the performance of groups with different social levels in neural circuits (mPFC-vHPC) and LFP recording in two different degrees of difficulty.”

Discussion “According to our findings, in this study, the subordinate rats showed better performance at the behavioral and electrophysiological levels compared to the other two groups. At the behavioral level, the subordinate rats were able to respond faster in choosing the target arm, as well as have a higher response accuracy, and at the level of electrophysiology, they had a higher theta correlation and coherence of mPFC-vHPC than the other two ranks. As the task becomes more difficult, the behavioral performance of the dominant rats remains unchanged, while the subordinate rats experience a decrease in performance. At the level of electrophysiology, in all three social groups, the coherence of the circuit decreases with the increases in the difficulty of the task. This shows that the dominant rats were able to maintain the same behavioral performance despite reduced mPFC-vHPC coherency”.

#Reviewer2

1. Given the dHPC's primary role in spatial working memory, alongside the vHPC, the authors should provide a more detailed explanation for their decision to focus solely on the ventral part of this brain region.

Answer: Although most studies have investigated the synchronization between PFC and dHPC, the synchronization between PFC and vHPC is stronger. For instance, O'Neill et al.'s study (2014) on rats showed that the input power of vHPC is very important for mPFC and plays a significant role in synchronizing the oscillatory activity of mPFC with dHPC area. In other words, it is the theta rhythm in the vHPC that regulates the synchrony between the other two regions [1,2]. Moreover, regarding working memory performance, the synchronization of vHPC-mPFC circuit plays a determining role.

In the manuscript changes are as follows.

#Introduction: Moreover, the intricate interplay between the mPFC and vHPC extends to their joint involvement in SWM tasks. The successful execution of SWM tasks relies on the coordinated activity of these regions, with the ventral hippocampus (vHPC) and dorsal hippocampus (dHPC) contributing to spatial encoding, and the mPFC facilitating executive control [16]. Notably, the synchronization of theta rhythms between the mPFC and vHPC fosters long-range connectivity crucial for SWM formation [17,18]. In this study, we used vHPC because the synchronization between PFC and vHPC is stronger than dHPC. Also, inputs from vHPC are very important for mPFC and plays a significant role in synchronizing the oscillatory activity of mPFC with dHPC area. In other words, theta rhythm in the vHPC regulates the synchrony between these two regions [19,20].

2. A more detailed explanation of the system used to determine the precise timing of the decision point is necessary. Did they use video-EEG?

Answer: In order to accurately extract the decision point time, videos were taken from all recorded moments and the times were extracted from videos. We described the details in data analysis section in method. We didn’t use video-EEG.

#Method: The process of extracting the data from the electrophysiology section was done by using videos recorded during the LFP signal recording, in order to match the time of behavioral activity with LFP signals. Then, the videos were analyzed and the decision point was extracted with millisecond precision. This time is actually the moment of turning the rat’s head towards the target for the last time, which three seconds before this time and one second after this moment (4 seconds in total) was considered as the decision point.

3. a) I have some questions about the T-maze task. Could the authors please provide a more detailed description of the protocol? Did they impose a time limit for the animals to complete the task? Were any animals excluded from the behavioral testing?"

Answer: we thank the esteemed

---

## [Decision Letter · Decision Letter 1]

31 Jan 2025

PONE-D-24-36279R1Social subordination is associated with better cognitive performance and higher theta coherence of the mPFC-vHPC circuit in male ratsPLOS ONE

Dear Dr. Heysieattalab,

Thank you for submitting your manuscript to PLOS ONE. After careful consideration, we feel that it has merit but does not fully meet PLOS ONE’s publication criteria as it currently stands. Therefore, we invite you to submit a revised version of the manuscript that addresses the points raised during the review process.

**ACADEMIC EDITOR: ** - please do make required changes 

We look forward to receiving your revised manuscript.

Kind regards,

Prof. Dr. Dragan Hrncic, MD, PhD

Academic Editor

PLOS ONE

Journal Requirements:

Reviewers' comments:

Reviewer's Responses to Questions

**Comments to the Author**

1. If the authors have adequately addressed your comments raised in a previous round of review and you feel that this manuscript is now acceptable for publication, you may indicate that here to bypass the “Comments to the Author” section, enter your conflict of interest statement in the “Confidential to Editor” section, and submit your "Accept" recommendation.

Reviewer #2: All comments have been addressed

2. Is the manuscript technically sound, and do the data support the conclusions?

Reviewer #2: Yes

3. Has the statistical analysis been performed appropriately and rigorously? 

Reviewer #2: Yes

4. Have the authors made all data underlying the findings in their manuscript fully available?

Reviewer #2: Yes

5. Is the manuscript presented in an intelligible fashion and written in standard English?

Reviewer #2: Yes

6. Review Comments to the Author

Reviewer #2: All the main issues have been addressed; in the Abstract the sentence "By increasing task difficulty, theta rhythm

coherency was reduced for all social ranks but subordinates maintained better task performance

with less reduction of theta coherence." should be reviewed and avoid the word 'coherency': Authors can use synchrony (or synchronization).

Also do not report "for the first time or similar in the abstract.

A further revision of the text to enhance its sound/readability would be appreciated.

7. PLOS authors have the option to publish the peer review history of their article (what does this mean?). If published, this will include your full peer review and any attached files.

Reviewer #2: No

---

## [Author Response · Author response to Decision Letter 2]

3 Feb 2025

#Journal Requirements:

Answer: Thanks for your comment, we ensure all of the references are complete and correct.

#Reviewer2

1. in the Abstract the sentence "By increasing task difficulty, theta rhythm coherency was reduced for all social ranks but subordinates maintained better task performance with less reduction of theta coherence." should be reviewed and avoid the word 'coherency': Authors can use synchrony (or synchronization).

Answer: Thanks for your comment, the mentioned item has been corrected in the text.

#Abstract: Although theta rhythm synchronization was reduced in all social ranks by increasing task difficulty, the subordinates maintained better task performance and less reduction of theta coherence.

2. Also do not report "for the first time or similar in the abstract.

Answer: we are grateful for your comment, it has been corrected.

#Abstract: These findings underscore the association between social hierarchy and working memory performance within the mPFC-vHPC circuit, highlighting the influence of social rank on implicated circuit.

3. A further revision of the text to enhance its sound/readability would be appreciated.

Answer: we thank the esteemed reviewer for making this point, we tried to make it readable.

#Abstarct: Social dominance hierarchy is considered an influential factor on cognitive performance. The spatial working memory (SWM) is inversely related to dominance status after the formation of social hierarchy. However, their neural underpinings are poorly understood. The medial prefrontal cortex (mPFC) and ventral hippocampus (vHPC) play pivotal roles in social hierarchy and SWM. To investigate the associations between social hierarchy and SWM and their neural circuit (mPFC-vHPC), we used twenty-one natal male Wistar rats after weaning (3 rats per cage, 7 cages in total). In the 9th postnatal week, the tube test was started to determine the relative social rank in each cage (dominant, middle-ranked, subordinate). One month after living in the hierarchy, we implanted electrodes in mPFC and vHPC. One week following recovery, the SWM test was performed using T-maze with two difficulty levels (30s and 5min delays between trials) while recording the local field potentials. The percentage of correct responses showed no significant difference among three different social groups. However, subordinates demonstrated significantly lower latency in reaching the goal arm, while middle-ranked rats exhibited the longest latency in 30s delay. Electrophysiological data revealed significantly higher theta correlation and coherence of the mPFC-vHPC circuit in subordinates. Although theta rhythm synchronization was reduced in all social ranks by increasing task difficulty, the subordinates maintained better task performance and less reduction of theta coherence. These findings underscore the association between social hierarchy and working memory performance within the mPFC-vHPC circuit, highlighting the influence of social rank on implicated circuit.

---

## [Editor Report · Decision Letter 2]

27 Feb 2025

Social subordination is associated with better cognitive performance and higher theta coherence of the mPFC-vHPC circuit in male rats

PONE-D-24-36279R2

Dear Dr. Heysieattalab,

We’re pleased to inform you that your manuscript has been judged scientifically suitable for publication and will be formally accepted for publication once it meets all outstanding technical requirements.

Kind regards,

Prof. Dr. Dragan Hrncic, MD, PhD

Academic Editor

PLOS ONE
---

## [Editor Report · Acceptance letter]

PONE-D-24-36279R2

PLOS ONE

Dear Dr. Heysieattalab,

I'm pleased to inform you that your manuscript has been deemed suitable for publication in PLOS ONE. Congratulations! Your manuscript is now being handed over to our production team.

Kind regards,

on behalf of

Professor Dragan Hrncic

Academic Editor

PLOS ONE